# Randomized-MLP Regularization Improves Domain Adaptation and Interpretability in DINOv2

**Joel Valdivia Ortega** [1,2,3,*]     **Lorenz Lamm** [1,3,4]     **Franziska Eckardt** [5]

**Benedikt Schworm** [5]     **Marion Jasnin** [2,6,*]     **Tingying Peng** [1,*]

[1]Helmholtz AI, Helmoltz Munich, Neuherberg, Germany
[2]Helmholtz Pioneer Campus, Helmholtz Munich, Neuherberg, Germany
[3]School of Computation, Information and Technology, TUM, Garching, Germany
[4]Biozentrum, University of Basel, Basel, Switzerland
[5]Department of Ophthalmology, LMU University Hospital, LMU Munich, Munich, Germany
[6]Department of Chemistry, TUM, Garching, Germany.

{joel.valdiviaortega, lorenz.lamm,
marion.jasnin, tingying.peng}@helmholtz-munich.de
{franziska.eckardt,benedikt.schworm}@med.uni-muenchen.de

## Abstract

Vision Transformers (ViTs), such as DINOv2, achieve strong performance across domains but often repurpose low-informative patch tokens in ways that reduce the interpretability of attention and feature maps. This challenge is especially evident in medical imaging, where domain shifts can degrade both performance and transparency. In this paper, we introduce Randomized-MLP (RMLP) regularization, a contrastive learning-based method that encourages more semantically aligned representations. We use RMLPs when fine-tuning DINOv2 to both medical and natural image modalities, showing that it improves or maintains downstream performance while producing more interpretable attention maps. We also provide a mathematical analysis of RMLPs, offering insights into its role in enhancing ViT-based models and advancing our understanding of contrastive learning.[1]

## 1 Introduction

Learning robust visual representations remains a central challenge in computer vision. Transformer-based models, such as Vision Transformers (ViTs) [11], have emerged as powerful backbones, especially when trained with self-supervised methods like contrastive or reconstruction-based learning [30, 5, 15, 32, 20]. These approaches yield generalizable models, which can be fine-tuned for specialized tasks, particularly in domains with limited labeled data, like medical and biological imaging [21, 19, 8, 25].

However, despite their empirical success, ViTs exhibit persistent issues in how they encode and distribute semantic information. Previous studies have observed that large ViTs often store global context within semantically weak or background regions [41, 38, 9]. These structural artifacts can

---

*Corresponding authors

[1]Code and pre-trained models are available at `https://github.com/peng-lab/rmlp`.

39th Conference on Neural Information Processing Systems (NeurIPS 2025).

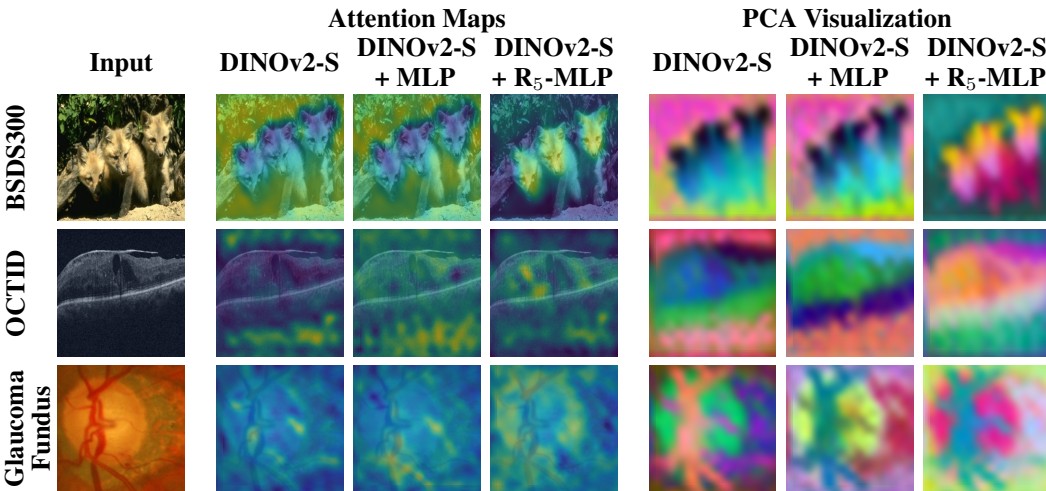

Figure 1: **Left:** Input image from BSDS300 [27], OCTID [14] and Glaucoma Fundus [2] datasets from top to bottom. **Middle:** Second- (BSDS300) and first-order (OCTID/Glaucoma Fundus) attention maps (see Section 1) over grayscale image. **Right:** Visualizations of top-3 PCA of patch tokens using DINOv2-S, DINOv2 fine-tuned on selected modality, and DINOv2 fine-tuned on selected modality using our regularizer. Colors are assigned in the RGB regime as the norm of the principal components. More qualitative results are available in Appendix A.1.

degrade performance on dense prediction tasks and compromise interpretability, a critical concern in biomedical domains [16, 46].

Our analysis reveals that these artifacts are not limited to large ViTs. Even though the small variant of DINOv2 [30] performs well on natural image tasks (Table 1), it still exhibits anomalous attention behavior. In medical imaging, DINOv2-S struggles to generalize to dense tasks such as segmentation (Table 2), despite achieving strong performance on classification tasks (Table 3a). This supports the hypothesis that it sacrifices local detail in favor of encoding global information through uninformative tokens. By analyzing patch token norms and their principal components, we identify *first-order artifacts*—where high-norm tokens align with background regions in ophthalmological images—and *second-order artifacts*—where the variance captured by the top three principal components is misaligned with semantic relevance in natural images (Figure 1). First order artifacts are more pronounced in medical images, where DINOv2-S statistically attends more to void or anatomically irrelevant regions than to biologically meaningful tissue (Table 4a).

To address these issues, we propose the *Randomized Multi-Layer Perceptron* (RMLP), a lightweight, theory-driven regularization module that improves semantic alignment in ViT representations. RMLP is motivated by the geometry of contrastive learning [5, 35] and enhances interpretability without requiring retraining or architectural changes. When applied to DINOv2, RMLP yields significant improvements in both natural and biomedical domains, achieving state-of-the-art classification and segmentation performance on ophthalmology datasets with minimal computational overhead. Additionally, we demonstrate that RMLP generalizes beyond DINOv2 by fine-tuning SwAV [4]—a model trained under a different self-supervised paradigm— and showing it consistently improves performance across modalities, indicating the broader applicability of our approach.

## 2 Related Work

**Artifacts in Transformer Representations.** Transformers are known to exhibit uneven attention allocation across input tokens. In NLP, Xiao, et al. [41] showed that early-position tokens receive disproportionate attention, regardless of their semantic importance. Sun, et al. [38] attributed such behavior to sparse, high-norm activations. Extending these observations to vision, Darcet, et al. [9] found that ViTs often produce a small set of high-norm patch tokens concentrated in background regions, which act as "registers" for global context. While these patch repurposing have been mostly studied in large-scale ViTs, our work shows it also emerge in smaller models like DINOv2-S.

**Mitigating Attention Artifacts.** Prior attempts to address attention artifacts have focused on heuristic or architectural solutions. Darcet, et al. [9] proposed adding learnable tokens and retraining the model on proprietary datasets. Jiang, et al. [18] showed that even untrained tokens can mitigate high-norm anomalies by absorbing excess activation. Others introduced auxiliary loss terms [40] or attention-smoothing modules [42] to encourage more uniform feature distributions. While these methods show empirical improvements, they lack theoretical grounding, require expensive retraining, or cannot be used as domain adaptation techniques. In contrast, our proposed RMLP module is easy to integrate, requires no retraining, and is grounded in the geometry of contrastive representation spaces. It also improves both semantic fidelity and interpretability across modalities.

## 3 Randomized-Multi-Layer Perceptron (RMLP)

Central to our approach is the observation that the structure of the representation heads—namely, the DINO [5] and iBOT [44] heads—play a key role in how information is encoded across patch and class tokens. In DINOv2, the contrastive framework aligns student and teacher global representations via the DINO head (operating on the class token), while the iBOT head introduces masked image modeling by aligning patch tokens through a cross-entropy objective. Both heads are implemented as MLPs.

Building on the findings of Darcet, et al. [9], we hypothesize that the representation heads are a key enabler of this behavior. To counteract this, we replace the learnable MLP heads with a randomized, non-trainable operator designed to preserve the topology of the representation space. This encourages the backbone to learn more robust and interpretable features, while preventing the heads from exploiting token-level classification shortcuts.

To avoid modifying the architecture, we first express the structure of the DINO and iBOT heads. A standard MLP [30] can be written as $f = \phi_r \circ \alpha_{r-1} \circ \cdots \circ \alpha_1 \circ \phi_1$, where $\phi_i$ are linear layers and $\alpha_i$ activations. We replace each $\phi_i$ with a randomized map $\varphi_i : \mathbb{R}^m \to \mathbb{R}^n$ as

$$\varphi_i((x_1, \ldots, x_m)) = (x_1, \ldots, x_n) + \Gamma_i(x_1, \ldots, x_m)^\top, \tag{1}$$

where $(x_1, \ldots, x_n)$ can be a truncated or zero-padded version of the input vector, $\Gamma_i \in \mathbb{R}^{n \times m}$ is a Gaussian matrix with i.i.d. entries drawn from $\mathcal{N}(0, \lambda/n)$ and $\lambda$ is a tunable *amplitude*. This operator can also be seen as a residual connection. Thus, we formally define an $R_\lambda$-MLP as

$$g = \varphi_r \circ \alpha_{r-1} \circ \cdots \circ \alpha_1 \circ \varphi_1. \tag{2}$$

RMLPs introduce no trainable parameters, yet remain fully compatible with end-to-end contrastive training objectives like DINO [5] and iBOT [44].

## 4 Theoretical Analysis

We now develop a topological framework to analyze how ViT embeddings evolve while passing through transformer blocks (Theorem 1), the way it generalizes from its training data into a bigger domain (Theorem 2 and Corollary 3) and how our RMLP regularizer impacts the contrastive learning paradigm (Theorem 4). This enables the characterization of RMLPs as random operators which turn point embeddings into *probability balls* (Corollary 5), improving robustness and promoting sparcity without altering the topology of learned representations when an adequate amplitude is chosen.

**Theorem 1.** *Let $\Omega$ be an image space and $\Psi$ a latent space. A ViT $\mathcal{V} : \Omega \to \Psi$ can be decomposed as a tokenization function $\mathcal{C} : \Omega \to \Psi$ followed by a sequence of transformer blocks $T : \Psi \to \Psi$, which are defined by local orthonormal bases generated in the attention heads by the queries, keys, and values layers along with the input data itself.*

*Proof.* ViTs use the attention mechanism to decompose the data using queries and keys layers, creating a field that data follows during the representation process. The value layers act as embeddings within the space defined by queries and keys. For a complete proof, refer to Theorem 11 in Appendix A.2. □

Training with the KoLeo regularizer [35] ensures that $\mathcal{V}$ remains injective since the loss, namely,

$$\mathcal{L}_{\text{KoLeo}}(\{x_1, \ldots, x_n\}) = -\frac{1}{n} \sum_{i=1}^{n} \log \left( \min_{j \neq i} \|x_i - x_j\|_2 \right)$$

would diverge otherwise. Continuity of $\mathcal{C}$, along with KoLeo and augmentations, guarantees that $T$ is locally injective (see Def. 6) on $\mathcal{C}[\Omega]$. These properties allow $T$ to generalize from its training data into $\Psi$ while preserving its embedding ability as shown in Corollary 3.

**Theorem 2.** *Being an homeomorphism an invertible continuous function with continuous inverse, assuming $T$ is locally injective and taking $\Psi$ to be a metric space, there exists $\varepsilon > 0$ such that $T$ is an homeomorphism on $\{x \in \Psi : \delta(x, \mathcal{C}[\Omega]) < \varepsilon\}$, where $\delta$ is the distance on $\Psi$.*

*Proof.* $T$ is continuous from Theorem 1. Since images can only take values from a finite set, $\Omega$ is bounded and by construction, $\mathcal{C}$ is continuous. Further, taking the codomain of $\mathcal{C}$ as a metric space and using again that images take discrete values, we can assume $\mathcal{C}[\Omega]$ is compact because of being a finite union of closed sets, namely singletons of images. Thus, for Lemma 12 in Appendix A.2, an $\varepsilon > 0$ exists such that $T$ is injective in $\{x \in \Psi : \delta(x, \mathcal{C}[\Omega]) < \varepsilon\}$, i.e., is invertible in that subset. Furthermore, since $\Psi$ is metric, the result follows from Lemma 13 in Appendix A.2. $\qquad\square$

**Corollary 3.** *If $\mathcal{A} \subseteq \Omega$ is the training data, and $T$ is locally injective on $\mathcal{A}$, the following holds:*

   ◍) *There exists $\varepsilon > 0$ such that $T$ is an homeomorphism on an $\varepsilon$-cloud containing $\mathcal{A}$ (see Def. 7).*

   ·) *Let $P, Q \subseteq \Omega$. $\mathcal{C}[P] \cup \mathcal{C}[Q]$ is disconnected in $\Psi$ if and only if $\mathcal{V}[P] \cup \mathcal{V}[Q]$ is disconnected in $\Psi$ (see Def. 8), which can contribute to the batch effect.*

   :) *If $D \subseteq \Psi$ is dense in $\Psi$ (see Def. 9), then $\mathcal{V}[D]$ is dense in $\mathcal{V}[\Psi]$.*

*Proof.* A complete proof is in Appendix A.2 in Theorem 18. $\qquad\square$

**Theorem 4.** *Let $\{p_1, \ldots, p_N\} \subseteq \mathbb{R}^m$, $\varepsilon > 0$, and $\lambda > 0$, with $\Gamma$ a matrix of size $n \times m$ with i.i.d. normal entries $\mathcal{N}(0, \lambda n^{-1})$. Then, for each $x \in \mathbb{R}^m$, $\mathbb{E}[\|\Gamma x\|_2^2] = m\lambda n^{-1}\|x\|_2^2$. Moreover, $\Gamma$ produces an $\varepsilon$-distortion (see Def. 14) on the set $E = \left\{ \frac{p_i - p_j}{\|p_i - p_j\|} : 1 \leqslant i < j \leqslant N \right\}$ with high probability if*

$$\lambda n^{-1} < \frac{\varepsilon^2}{8 \ln N}.$$

*Proof.* (*Sketch*) Using concentration results for Gaussian matrices, we follow [28]. A complete proof can be found at Appendix A.2 in Theorem 19. $\qquad\square$

**Corollary 5.** *If $E = \{x_1, \ldots, x_N\} \subseteq \mathbb{S}^{m-1}$, $\varepsilon > 0$, and $\varphi : \mathbb{R}^m \to \mathbb{R}^n$ is defined as in Eq. 1 with $\lambda n^{-1} < \varepsilon^2/(8 \ln N)$, then for all $i \in \{1, \ldots, N\}$,*

$$\|(x_{i,1}, \ldots, x_{i,n}) - \varphi(x_i)\|_2 < \varepsilon$$

*with high probability.*

*Proof.* This follows directly from the definition of $\varphi$ and Theorem 4. $\qquad\square$

We now analyze the impact of RMLP heads on image embeddings. Let us take an RMLP $g$ as in Equation 2 with amplitude $\lambda$ and GELU activation acting on the embedding $\mathcal{V}(x) = ((z_{i,j})_{j \leqslant d})_{i \leqslant \nu}$. Applying Corollary 5, the first random layer $\varphi_1$ turns each token $z_i$ into a ball of radius $\varepsilon$, determined by the embedding's dimension, $\lambda$, and the number of tokens $\nu$.

By Theorem 4, the output of $\varphi_1(\mathcal{V}(x))$ lies on a sphere of radius $d\lambda n_1^{-1}$ on expectation. The contrastive losses encourage similarity between views, while the KoLeo regularizer promotes injectivity. However, the GELU activation restricts the space to a spherical region with mostly positive coordinates, where dispersion is optimized via approximate orthogonality among embeddings. Successive layers $\varphi_2, \varphi_3, \varphi_4$ expand the radius of probability balls further. Thus, $g \circ \mathcal{V}(x)$ becomes a point cloud creating a probability ball, rather than individual points, regularizing the learning by applying cross-entropy over neighborhoods. If the RMLP amplitude $\lambda$ is too large, the topology of the embedding space may be distorted, as shown in Figure 2. Remarkably, the topological properties of ViTs remain unaffected whether MLPs or RMLPs heads are used during training, as our analysis is independent of it.

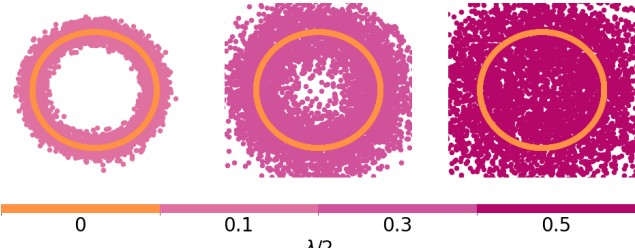

Figure 2: Visualization of a randomized operator following Equation 1, applied to the unit circle in $\mathbb{R}^2$ ($\lambda/2 = 0$) for different values of $\lambda$. For $\lambda/2 = 0.1$, the point cloud retains a circular structure, preserving the topology of the circle. For $\lambda/2 = 0.5$, the point cloud no longer exhibits a circular topology, as it transitions to a disk-like shape.

# 5 Implementation and Training Details

**Model Variants.** We use two ViT-based self-supervised models: DINOv2-S [30] and SwAV [4]. Fine-tuned variants include standard MLP heads and our regularized version using $R_\lambda$-MLP. Fine-tuning follows the DINOv2 protocol with the original head architecture retained. Hyperparameters and external code can be found in Tables 5, 6.

**Training Modalities.** Each backbone is fine-tuned separately on three modalities: ImageNet-1k (natural), Colour Fundus Photography (CFP), and Optical Coherence Tomography (OCT) (Table 7a). Medical datasets are sourced from public subsets aggregated in RETFound [43], ensuring geographic diversity (Table 7b).

**Hybrid Architecture for Dense Tasks.** For pixel-level predictions, we pair the ViT encoder with a UNet-style decoder [33, 17]. Patch and class tokens are projected and fused with early UNet features, enabling multi-scale feature integration. This ViT-UNet hybrid is used for segmentation and depth estimation tasks.

# 6 Experimental Setting

**Evaluation Setup.** We fine-tune DINOv2-S and SwAV with $R_\lambda$-MLP at amplitudes {0.1, 5, 10, 20}, running 10 trials per modality and setting. Baselines use MLP heads. Downstream tasks employ either linear probes or the hybrid decoder depending on task type. Results are averaged over runs and tested for significance using the Mann–Whitney U test [26], appropriate under our distributional assumptions. Qualitative results for both natural and ophthalmological modalities can be found in Appendix A.1.

**Natural Image Tasks.** On ImageNet-1k [34], we perform image classification training with cross-entropy. For semantic segmentation of [45] and depth estimation of [36], we apply a linear head and the ViT-UNet hybrid. Segmentation is trained with a linear combination of focal [23] and dice loss [29]. Depth maps are rescaled to [0, 50] and trained with focal loss. Table 1 reports performance compared to baselines [9, 40, 42].

**OCT and CFP Tasks.** We assess disease classification and retinal layer segmentation. Segmentations emphasize the ophthalmologically relevant Outer Nuclear Layer (ONL). UNet output biases are initialized (-2 for background, 2 for others) for stable training. We benchmark against RETFound [43] and a domain-specific model [12] (Tables 2, 3).

**Cross-Modal Attention Analysis.** We visualize attention maps across modalities: second-order for natural images [27] and first-order for OCT [14] and CFP [2]. High vs. low-information patches are separated using a 2-component Gaussian Mixture Model on smoothed gradient at patch level.

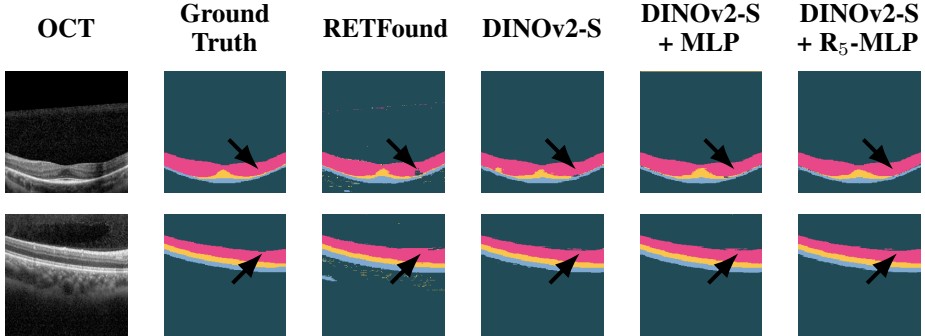

Figure 3: Semantic segmentation of retinal layers on the dataset from Eckardt, et al. [12] (OCT) using different backbones (columns) in a ViT-UNet hybrid. Black arrows highlight cases where fine-tuning DINOv2-S with $R_5$-MLP outperforms other models.

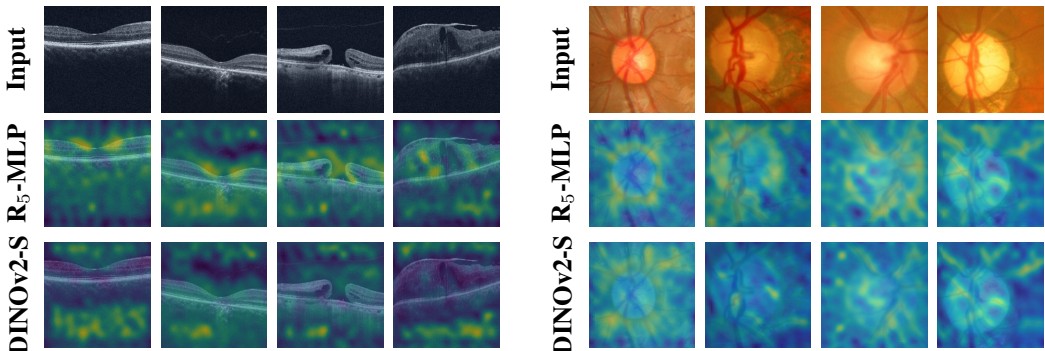

Figure 4: **Top**: OCTID [14] (left) and Glaucoma Fundus [2] (right) images. **Middle**: First-order attention maps from model fine-tuned with $R_5$-MLP overlaid on grayscale version of input image. **Bottom**: First-order attention maps from DINOv2-S overlaid on grayscale version of input image. Brighter maps indicate more attention. While DINOv2-S focuses on void regions, DINOv2-S with $R_5$-MLP attends to key anatomical features in the OCTs—such as the foveal pit, macular hole, and epithelial detachment—and highlights the rim and blood vessels in Fundus images.

## 7  Results and Discussion

**Effectiveness of RMLP Regularization.** Across natural and medical image domains, fine-tuning with $R_\lambda$-MLP consistently outperforms MLP baselines. On natural images, it better preserves performance compared to other regularizers. In the medical domain, despite limited data and single-GPU training, $R_\lambda$-MLP achieves state-of-the-art OCT segmentation (Figure 3) and improves classification, especially with 1-NN, indicating enhanced representation quality (Table 3). While linear-head classification with $R_\lambda$-MLP slightly trails state-of-the-art or DINOv2-S, this reflects differing latent space structures since RMLPs promote neighborhood-based similarity (Theorem 4, Corollary 5), whereas MLPs enforce linear separability. Larger $\lambda$ values degrade performance by altering representation topology (Tables 1–3, Figure 2), while smaller amplitude values do not consistently improve downstream results, highlighting a tradeoff between retaining information and preserving topological structure ViTs need to overcome during fine-tuning.

**Enhancement on Cross-Paradigm Learning.** RMLPs enable effective adaptation of models trained under different learning paradigms, such as SwAV [4], when combined with DINOv2 learning algorithm (Tables 2b, 3b,). While applying DINOv2 to standard MLPs also yields gains, RMLPs consistently deliver superior performance across tasks.

Table 1: Performance across downstream tasks on natural images. Semantic segmentation on ADE20k [45], depth estimation on NYU-Depth V2 [36] and image classification on ImageNet-1k [34]. Bold indicates best mean, † second-best on for each base model (DINOv2-S, DINOv2-G and SwAV). Underlined values are significantly better than DINOv2-S ($p < 0.001$); * and *** denote significance over base model fine-tuned with MLP at $p < 0.1$ and $p < 0.001$ (Mann–Whitney U test [26]). Results show mean±std from ten independently fine-tuned backbones.

| Model | Semantic Segmentation ↑ | | Depth Estimation ↓ | | Image Classification ↑ | | |
| --- | --- | --- | --- | --- | --- | --- | --- |
| | ViT-UNet Hybrid | Linear Head | ViT-UNet Hybrid | Linear Head | 1-Nearest Neighbor | Random Forest | Linear Head |
| | | | | DINOv2-S | | | |
| DINOv2-S [30] | **0.81±0.1** | †0.76±0.1* | **7±3***\*** | **9±3***\*** | **0.70***\*** | **0.18***\*** | **0.77±0.01***\*** |
| DINOv2-S + MLP | 0.78±0.1 | 0.75±0.1 | †8±3 | †10±3 | 0.64±0.01 | 0.14±0.01 | 0.73±0.01 |
| DINOv2-S + $R_{0.1}$-MLP | 0.61±0.1 | 0.66±0.1 | 8±1 | 9±1 | 0.64±0.01 | 0.15±0.01 | 0.74±0.06 |
| DINOv2-S + $R_5$-MLP | †0.80±0.1*** | **0.77±0.1***\*** | 7±3*** | 9±3*** | †0.65±0.01*** | †0.16±0.01*** | †0.76±0.01*** |
| DINOv2-S + $R_{10}$-MLP | †0.80±0.1*** | **0.77±0.1***** | 7±3*** | 9±3*** | †0.65±0.01*** | 0.15±0.01*** | **0.77±0.01***** |
| DINOv2-S + $R_{20}$-MLP | †0.76±0.2 | **0.77±0.1***** | 7±3*** | 9±3*** | 0.64±0.01*** | 0.14±0.01*** | 0.75±0.01*** |
| Registers [9] | 0.72±0.2 | 0.67±0.1 | 7±1 | 9±1 | 0.28 | 0.05 | 0.74±0.01** |
| DVT [42] | 0.45±0.2 | 0.64±0.01 | 7±1 | 9±1 | **0.7** | 0.11 | 0.64±0.01 |
| | | | | DINOv2-G | | | |
| Sinder [40] | 0.51±0.2 | 0.56±0.1 | 6±3 | 8±1 | 0.78 | 0.12 | †0.76±0.01*** |
| | | | | SwAV | | | |
| SwAV [4] | **0.37±0.1** | – | **12±1** | – | – | – | **0.69±0.01** |
| SwAV + MLP | 0.29±0.08 | – | †13±1 | – | – | – | **0.69±0.01** |
| SwAV + $R_5$-MLP | 0.34±0.09 | – | †13±2 | – | – | – | **0.69±0.01** |
| SwAV + $R_{10}$-MLP | 0.27±0.1 | – | **12±1** | – | – | – | **0.69±0.01** |
| SwAV + $R_{20}$-MLP | †0.36±0.1 | – | †13±1 | – | – | – | **0.69±0.01** |

Table 2: Performance metrics for semantic segmentation on Eckardt, et al. [12] dataset using a ViT-UNet hybrid. Results show mean±std of DICE scores from ten independently fine-tuned backbones using DINOv2 and SwAV as base models. Bold numbers indicate best performance, † the second-best, and underlined results outperform RETFound (p < 0.001, Mann-Whitney U test). Statistical significance vs. base model fine-tuned with MLP is shown by * (p < 0.1).

(a) Base model: DINOv2.

| Model | Averaged DICE | DICE on ONL |
| --- | --- | --- |
| RETFound [43] | 0.92±0.06 | 0.59±0.10 |
| Registers [9] | 0.92±0.1 | 0.62±0.21 |
| Sinder [40] | 0.90±0.21 | 0.61±0.2 |
| DVT [42] | 0.78±0.25 | 0.43±0.22 |
| DINOv2-S [30] | 0.79±0.20 | 0.54±0.20 |
| DINOv2 + MLP | 0.86±0.20 | 0.55±0.20 |
| DINOv2 + $R_{0.1}$-MLP | †0.94±0.12 | †0.68±0.21 |
| DINOv2 + $R_5$-MLP | **0.97±0.03*** | **0.72±0.09*** |
| DINOv2 + $R_{10}$-MLP | 0.87±0.20 | 0.61±0.20 |
| DINOv2 + $R_{20}$-MLP | 0.70±0.30 | 0.47±0.20 |
| Eckardt, et al. [12] | 0.92±0.03 | 0.44±0.03 |

(b) Base model: SwAV.

| Model | Averaged DICE | DICE on ONL |
| --- | --- | --- |
| SwAV [4] | †0.89±0.15 | 0.59±0.14 |
| SwAV + MLP | 0.88±0.15 | †0.6±0.18 |
| SwAV + $R_{0.1}$-MLP | **0.92±0.12** | **0.64±0.1** |
| SwAV + $R_5$-MLP | 0.83±0.27 | 0.58±0.2 |
| SwAV + $R_{10}$-MLP | 0.71±0.33 | 0.42±0.23 |
| SwAV + $R_{20}$-MLP | 0.84±0.26 | 0.58±0.21 |

**Attention Artifacts on Natural Images.** Fine-tuning on ImageNet-1k slightly reduces the CorLoc score for both MLP and RMLP heads (Table 4b), a minor drop consistent with other regularization methods and attributable to fine-tuning rather than the RMLPs. While regularized models from literature maintain global context and classification accuracy comparable to DINOv2-S, they degrade patch token quality in dense prediction tasks (Table 1). Notably, DINOv2-S encodes global information in patch tokens from low-information regions, a tendency amplified by regularized models. Fine-tuning with MLPs or RMLPs reduces this artifact, equalizing patch token behavior, with RMLPs better preserving downstream performance post fine-tuning (Table 4).

Table 3: Performance metrics for pathology classification across OCT and CFP modalities. Results show mean±std from ten independently fine-tuned backbones. Bold numbers indicate best performance, † the second-best, and underlined results outperform RETFound (p < 0.001, Mann-Whitney U test). Statistical significance vs. base model fine-tuned with MLP is shown by * (p < 0.1), ** (p < 0.01), and *** (p < 0.001).

(a) Accuracy for pathology classification using DINOv2-S as base model.

| Dataset | | DINOv2-S [30] | DINOv2-S + MLP | DINOv2-S + $R_{0.1}$-MLP | DINOv2-S + $R_5$-MLP | DINOv2-S + $R_{10}$-MLP | DINOv2-S + $R_{20}$-MLP |
|---|---|---|---|---|---|---|---|
| | | | | 1-Nearest Neighbor classification | | | |
| OCTID [14] | (0.8) | †0.75*** | 0.69±0.02 | 0.68±0.01 | **0.78±0.01***** | 0.73±0.01*** | 0.68±0.01*** |
| Glaucoma Fundus [2] | (0.78) | 0.67*** | 0.64±0.01 | 0.65±0.01 | **0.71±0.09***** | †0.69±0.09*** | 0.63±0.07 |
| IDRID [31] | (0.44) | 0.42*** | 0.43±0.03 | 0.36±0.01 | **0.47±0.01***** | †0.44±0.02*** | 0.43±0.04* |
| JSIEC [7] | (0.45) | 0.59*** | 0.51±0.02 | 0.51±0.01 | **0.67±0.01***** | †0.65±0.07*** | 0.59±0.08*** |
| MESSIDOR-2 [1, 10] | (0.56) | †0.53*** | 0.47±0.01 | 0.43±0.01 | **0.55±0.01***** | †0.53±0.08*** | 0.52±0.08*** |
| PAPILA [22] | (0.63) | †0.71*** | 0.65±0.03 | 0.65±0.01 | **0.74±0.02***** | †0.71±0.01*** | 0.65±0.03** |
| Retina [6] | (0.5) | 0.54 | 0.54±0.02 | 0.39±0.01 | **0.58±0.01***** | **0.58±0.01***** | †0.57±0.01*** |
| | | | | Random Forest classification | | | |
| OCTID [14] | (0.81) | **0.75***** | 0.54±0.01 | 0.58±0.01 | †0.74±0.01*** | 0.73±0.02*** | 0.69±0.02*** |
| Glaucoma Fundus [2] | (0.73) | **0.69***** | 0.65±0.08 | 0.67±0.01 | †0.68±0.08*** | 0.67±0.08** | †0.68±0.09** |
| IDRID [31] | (0.4) | †0.49*** | 0.46±0.01 | 0.47±0.01 | 0.47±0.01*** | 0.46±0.01** | **0.5±0.05***** |
| JSIEC [7] | (0.28) | 0.28*** | 0.26±0.08 | 0.25±0.01 | **0.32±0.08***** | †0.29±0.06*** | 0.28±0.06*** |
| MESSIDOR-2 [1, 10] | (0.58) | **0.58** | 0.58±0.01 | 0.58±0.01 | **0.58±0.01***** | **0.58±0.01***** | **0.58±0.01** |
| PAPILA [22] | (0.69) | **0.72***** | 0.68±0.01 | 0.68±0.01 | †0.71±0.01*** | †0.71±0.01*** | 0.69±0.09** |
| Retina [6] | (0.59) | **0.59***** | 0.55±0.07 | 0.55±0.01 | **0.59±0.07***** | †0.58±0.06*** | †0.58±0.08*** |
| | | | | Linear classification | | | |
| OCTID [14] | (0.93±0.05) | **0.88±0.08**** | 0.76±0.02 | 0.82±0.01 | †0.87±0.01*** | †0.87±0.01*** | 0.86±0.01*** |
| Glaucoma Fundus [2] | (0.83±0.06) | **0.79±0.06***** | 0.71±0.09 | 0.73±0.02 | 0.75±0.01*** | 0.75±0.01*** | †0.76±0.01*** |
| IDRID [31] | (0.44±0.04) | **0.50±0.04**** | †0.44±0.03 | 0.48±0.03 | 0.42±0.03 | 0.40±0.03 | †0.44±0.03 |
| JSIEC [7] | (0.72±0.01) | 0.73±0.07*** | 0.63±0.01 | 0.68±0.02 | **0.78±0.01***** | †0.76±0.01*** | 0.73±0.03*** |
| MESSIDOR-2 [1, 10] | (0.56±0.03) | 0.46±0.08 | †0.52±0.03 | 0.51±0.04 | **0.54±0.07*** | †0.52±0.08 | 0.49±0.09 |
| PAPILA [22] | (0.67±0.07) | **0.71±0.05** | 0.66±0.04 | 0.66±0.03 | †0.69±0.03 | 0.7±0.03 | †0.69±0.03 |
| Retina [6] | (0.51±0.03) | 0.52±0.04 | 0.53±0.02 | 0.51±0.02 | †0.54±0.03 | †0.54±0.03 | **0.55±0.02*** |

(b) Accuracy for pathology classification using SwAV as base model.

| Dataset | | SwAV [4] | SwAV+ MLP | SwAV+ $R_{0.1}$-MLP | SwAV+ $R_5$-MLP | SwAV+ $R_{10}$-MLP | SwAV+ $R_{20}$-MLP |
|---|---|---|---|---|---|---|---|
| OCTID [14] | (0.93±0.05) | **0.84±0.02** | †0.83±0.02 | **0.84±0.02** | **0.84±0.02** | 0.82±0.02 | **0.84±0.02** |
| Glaucoma Fundus [2] | (0.83±0.06) | **0.76±0.01** | **0.76±0.01** | †0.75±0.01 | **0.76±0.01** | **0.76±0.01** | **0.76±0.02** |
| IDRID [31] | (0.44±0.04) | 0.45±0.03 | **0.49±0.02** | 0.47±0.02 | **0.49±0.04** | 0.47±0.02 | †0.48±0.04 |
| JSIEC [7] | (0.72±0.01) | **0.73±0.01** | †0.72±0.02 | †0.72±0.01 | **0.73±0.01** | **0.73±0.01** | †0.72±0.01 |
| MESSIDOR-2 [1],[10] | (0.56±0.03) | **0.55±0.04** | **0.55±0.03** | **0.55±0.02** | †0.54±0.02 | **0.55±0.02** | **0.55±0.01** |
| PAPILA [22] | (0.67±0.07) | 0.63±0.05 | †0.65±0.04 | 0.62±0.03 | **0.66±0.04** | †0.65±0.04 | 0.64±0.04 |
| Retina [6] | (0.51±0.03) | 0.52±0.03 | †0.53±0.03 | †0.53±0.02 | **0.54±0.02** | 0.52±0.02 | **0.54±0.02** |

**Attention Artifacts on Ophthalmological Modalities.** Both RETFound and DINOv2-S, along with its regularized variants from literature, achieve strong classification accuracy on ophthalmology datasets using patch tokens alone (Table 4c), even from void regions, indicating excessive global information leakage. Moreover, these models exhibit weak or negative correlation with retinal layer presence (Table 4a, Figure 4). In contrast, models fine-tuned with RMLPs (using suitable $\lambda$) show better anatomical alignment and reduced attention artifacts as well as better performance in dense tasks (Table 2). This is crucial, as interpretable pathology detection relies on attending to anatomically relevant regions.

**ViT-UNet Hybrids.** Combining ViT backbones with UNet-style decoders consistently improves performance on dense prediction tasks (Tables 1, 8), highlighting the benefit of integrating global context with local spatial detail.

Table 4: Evaluation of patch tokens repurposing.

(a) Pearson correlation between patch tokens on the top 25% ranked by norm and presence of retinal layers in Eckardt, et al. [12]

| Model | Correlation |
|---|---|
| RETFound [43] | 0.0±0.01 |
| Registers [9] | -0.19±0.13 |
| Sinder [40] | 0.0±0.17 |
| DVT [42] | -0.37±0.09 |
| DINOv2-S [30] | -0.13±0.14 |
| DINOv2 + MLP | 0.19±0.03 |
| DINOv2 + $R_{0.1}$-MLP | 0.15±0.02 |
| DINOv2 + $R_5$-MLP | **0.21±0.03**[**] |
| DINOv2 + $R_{10}$-MLP | 0.2±0.04 |
| DINOv2 + $R_{20}$-MLP | 0.16±0.01 |

(b) CorLoc scores applying LOST [37] with default parameters on VOC07 [13] and accuracy on ImageNet-1k using 1-Nearest Neighbors on class tokens and patch tokens of minimum/maximum norm of second-order attention maps from low and high information patches respectively (see Section 1). Bold and underline indicate best and second-best accuracy per model.

| Model | CorLoc ↑ | Classification accuracy ↑ |
|---|---|---|
| Registers [9] | 35.20 | 0.71/0.38/0.34 |
| Sinder [40] | **33.97** | **0.78**/0.39/**0.37** |
| DVT [42] | 31.23 | 0.71/**0.42/0.37** |
| DINOv2-S [30] | 34.18 | 0.71/0.31/0.26 |
| DINOv2-S + MLP | 31.18±0.03 | 0.63±0.01/0.20±0.01/0.18 ±0.01 |
| DINOv2-S + $R_{0.1}$-MLP | 31.20±0.04 | 0.63±0.01/0.20±0.01/0.18 ±0.01 |
| DINOv2-S + $R_5$-MLP | 31.17±0.05 | 0.64±0.01/0.21±0.01/0.19±0.01 |
| DINOv2-S + $R_{10}$-MLP | 31.17±0.05 | 0.64±0.01/0.21±0.01/0.19±0.01 |
| DINOv2-S + $R_{20}$-MLP | 31.19±0.07 | 0.63±0.01/0.20±0.01/0.18±0.01 |

(c) Pathology classification accuracy on Eckardt, et al. [12], OCTID [14], and aggregated CFP datasets ([2, 31, 7, 1, 10, 22, 6]) using Random Forest with class tokens and patch tokens of minimum/maximum norm from low and high information patches respectively (see Section 1). Bold and underline indicate best and second-best accuracy per model per dataset.

| Model | Eckardt, et al. [12] | OCTID [14] | CFP |
|---|---|---|---|
| RETFound [43] | **1.0**/0.92 /0.87 | **0.77**/0.45/0.47 | **0.53**±0.17/0.51±0.17/0.51±0.17 |
| Registers [9] | 0.83/**0.87**/0.86 | **0.67**/0.55/0.62 | **0.54**±0.15/0.5±0.16/0.5±0.17 |
| Sinder [40] | 0.76/**0.8**/0.67 | **0.78**/0.56/0.55 | **0.54**±0.16/0.5±0.16/0.48±0.17 |
| DVT [42] | 0.77/0.7/**0.94** | **0.76**/0.61/0.57 | **0.55**±0.14/0.51±0.15/0.52±0.15 |
| DINOv2-S [30] | 0.74/**0.84**/0.77 | **0.73**/0.48/0.49 | **0.55**±0.14/0.49±0.17/0.49±0.18 |
| DINOv2 + MLP | **0.94**±0.03/0.67±0.07/0.86±0.02 | 0.55±0.02/0.36±0.01/0.38±0.01 | 0.53±0.15/0.47±0.17/0.47±0.17 |
| DINOv2-S + $R_{0.1}$-MLP | **0.95**±0.02/0.68±0.08/0.87±0.03 | 0.55±0.02/0.36±0.01/0.38±0.02 | 0.53±0.15/0.47±0.17/0.47±0.17 |
| DINOv2-S + $R_5$-MLP | **0.96±0.02**/ 0.67±0.07/0.86±0.02 | 0.55±0.02/0.36±0.01/0.37±0.01 | 0.53±0.15/0.47±0.17/0.48±0.17 |
| DINOv2-S + $R_{10}$-MLP | **0.94**±0.02/0.69±0.08/0.85±0.04 | 0.55±0.02/0.36±0.01/0.38±0.01 | 0.53±0.15/0.47±0.17/0.48±0.17 |
| DINOv2-S + $R_{20}$-MLP | **0.95±0.02**/0.7±0.07/0.87±0.02 | 0.55±0.01/0.36±0.01/0.37±0.01 | 0.53±0.15/0.47±0.17/0.47±0.17 |

# 8 Conclusions

Our work shows that RMLP regularization enhances the interpretability and robustness of ViT representations across both natural and ophthalmological domains. By inducing sparsity in patch tokens and mitigating first- and second-order artifacts, fine-tuning using $R_\lambda$-MLPs produces more structured embeddings. Despite being trained on limited data and using small computational resources, $R_\lambda$-MLPs help ViTs achieve strong performance in classification and dense prediction tasks both on natural and ophthalmological images. These results highlight randomized-MLPs as a lightweight and effective approach for regularizing representation geometry, pointing to a promising direction for developing more semantically grounded and interpretable vision transformers.

**Limitations.** The optimal regularization strength may depend on the data modality or the dimensionality of the ViT's latent space. However, this work does not propose a principled method for selecting it, relying instead on heuristic tuning. Additionally, our experiments are currently limited to small- and mid-scale datasets. Future work should investigate performance at larger scales and across a broader range of domains.

## Acknowledgments and Disclosure of Funding

J.V.O. and L.L. received support from the Helmholtz Association under the joint research school "Munich School for Data Science - MUDS". T.P and B.S. received support from the DFG grant 513025799. We want to thank Salome Kazeminia and Sophia J. Wagner for the valuable inputs and conversations during the making of this paper.

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

# A Technical Appendices and Supplementary Material

## A.1 Supplementary Figures

Figure 5 shows fine-tuning with RMLPs create tokens with smaller norm and that it assigns bigger norms to patch tokens coming from high-information regions unlike when fine-tuning with MLPs.

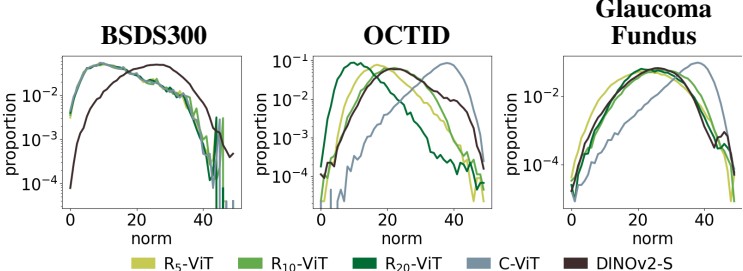

(a) Proportion curves of attention maps' norms.

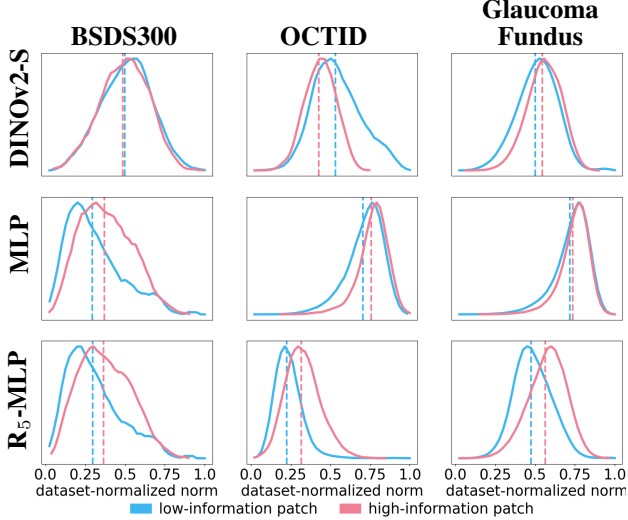

(b) Normalized probability density functions of attention maps' norms for low- and high-information patches across models (rows) and datasets (columns). Low- and high-content patches are classified by a two-component Gaussian Mixture model fitted on smoothed pixel gradients averaged per patch. Dashed lines stand for expected value.

Figure 5: Visualization of second- (BSDS300 [27]) and first-order (OCTID [14]/Glaucoma Fundus [2]) attention maps' norm statistics. **Subfigure (a):** Proportion curves. **Subfigure (b):** Normalized probability density functions for low- and high-information patches.

The following visualizations for the first- and second-order attention maps as well as the PCA visualizations were computed following a sliding window approach.

Here we present some extra examples on the performance of DINOv2-S, DINOv2-S+MLP and DINOv2-S+$R_\lambda$-MLP in natural, OCT and CFP modalities, in addition of RETFound [43] for the ophthalmology modalities mentioned before.

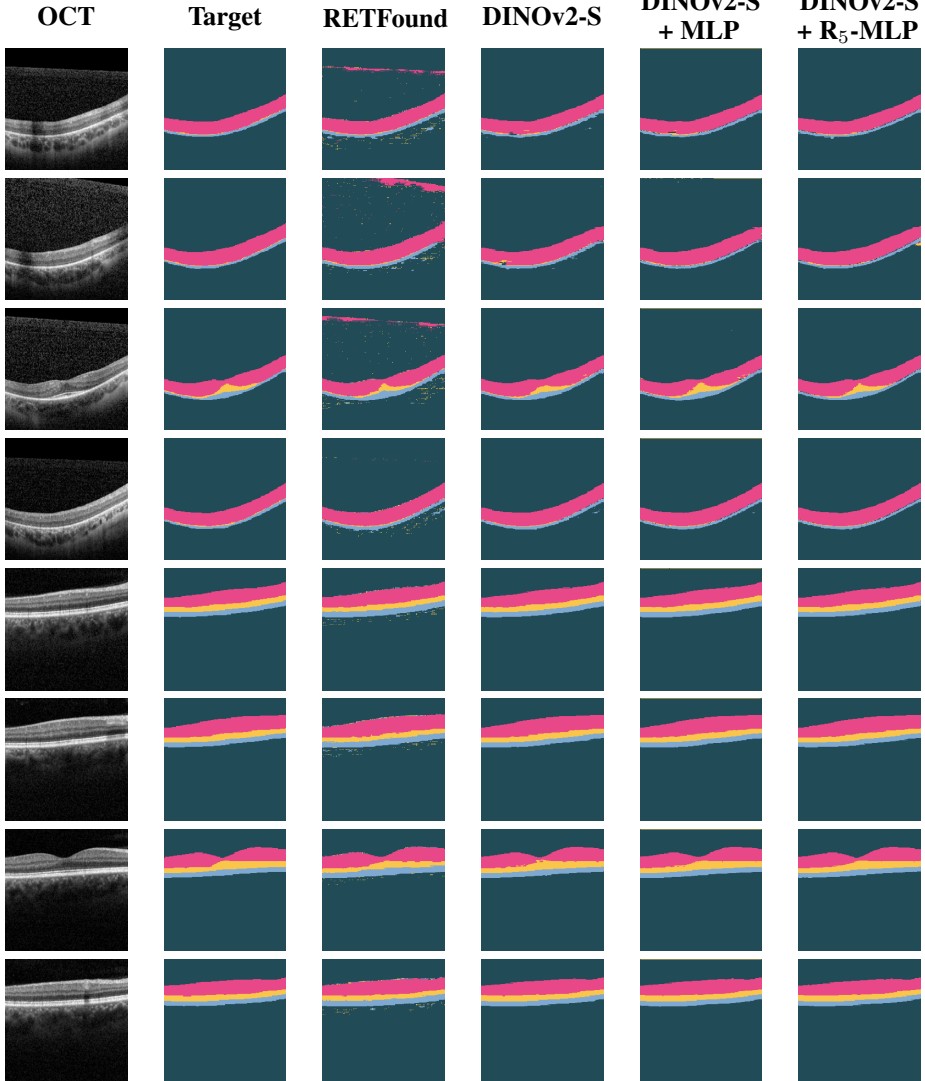

Figure 6: OCT segmentation on the dataset from Eckardt, et al. [12] using different backbones (columns) in a ViT-UNet hybrid

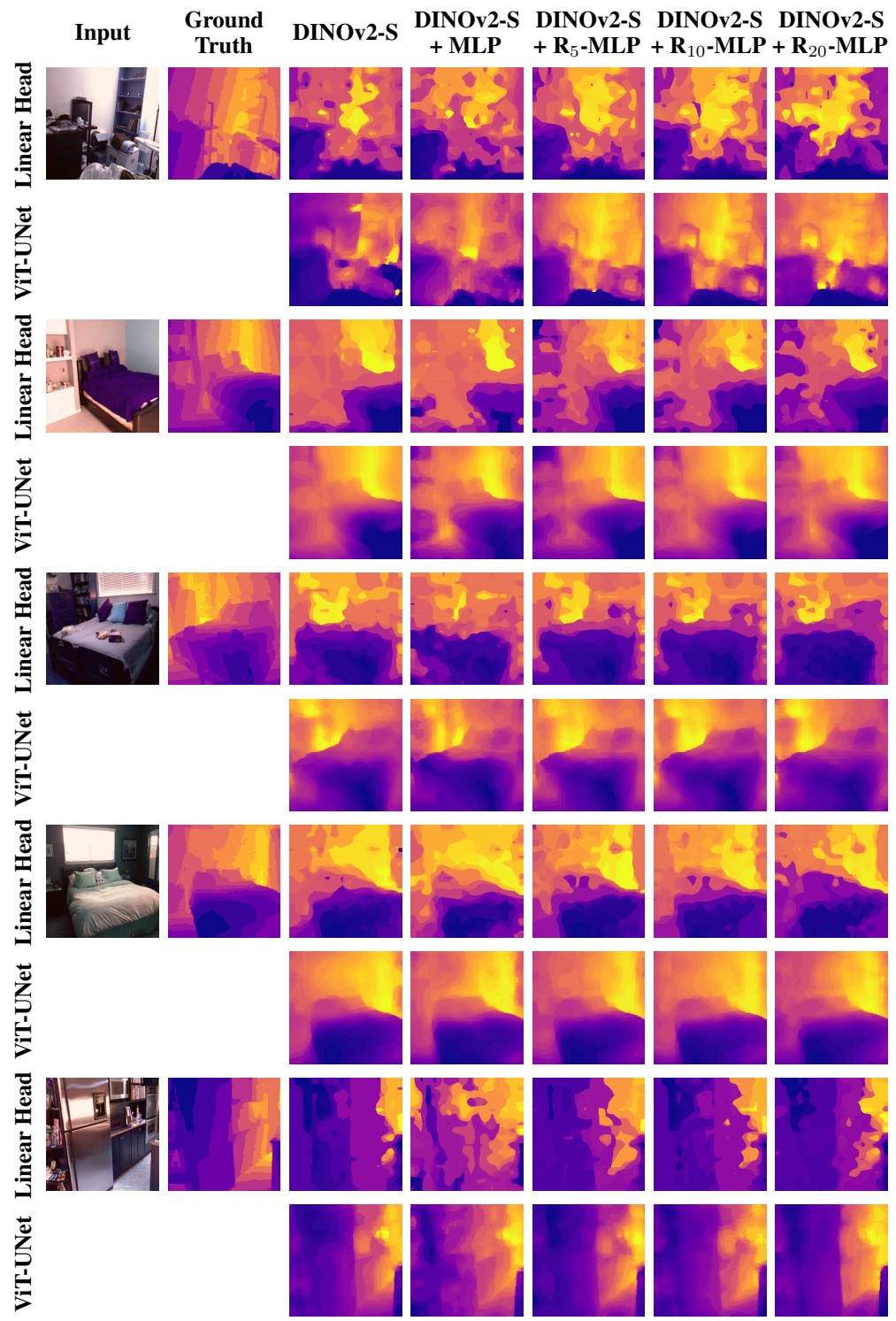

Figure 7: Depth estimation of NYU-Depth V2 [36] dataset using various backbones (last five columns) with a linear head or ViT-UNet hybrid.

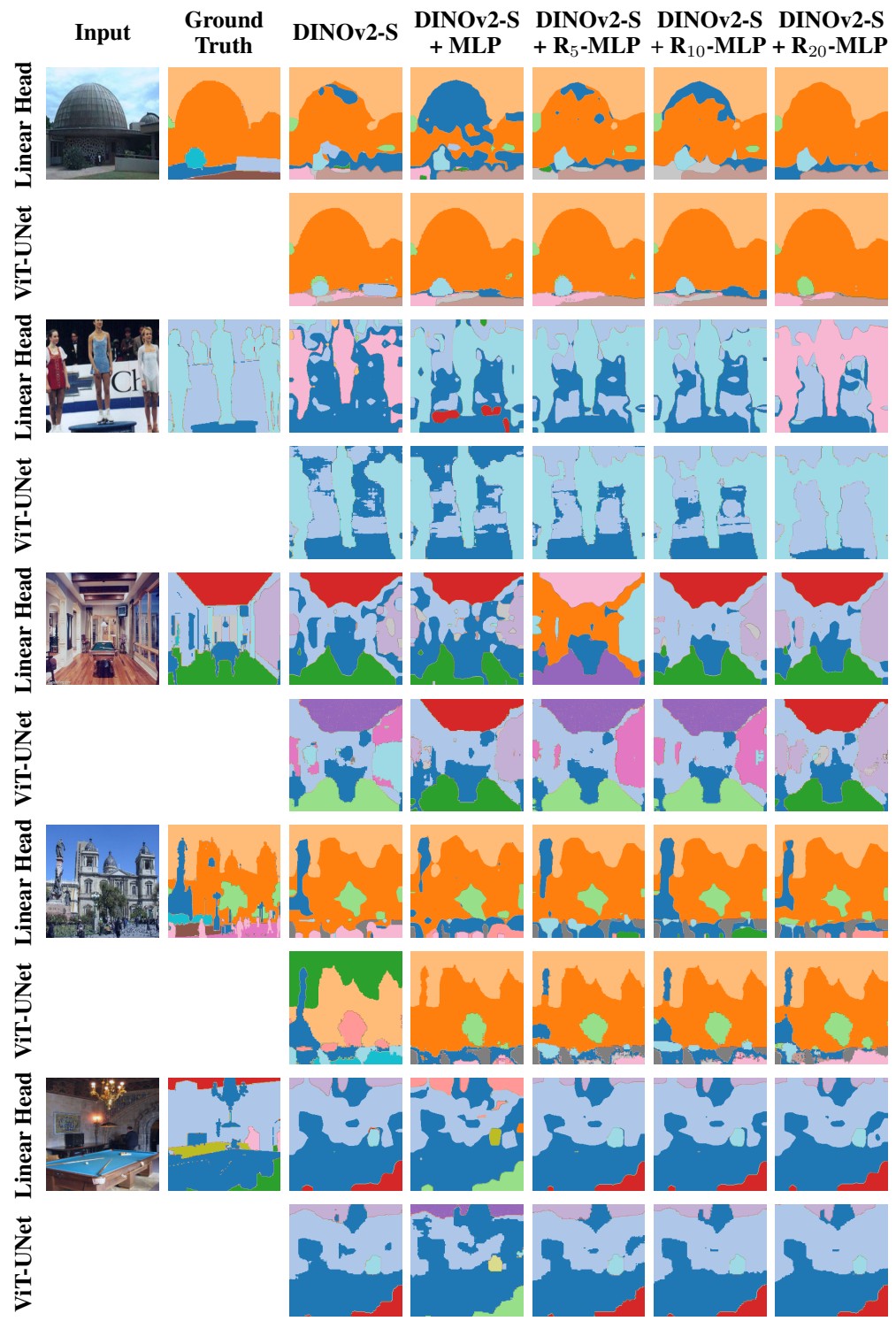

Figure 8: Semantic segmentation of ADE20k [45] dataset using various backbones (last five columns) with a linear head or ViT-UNet hybrid.

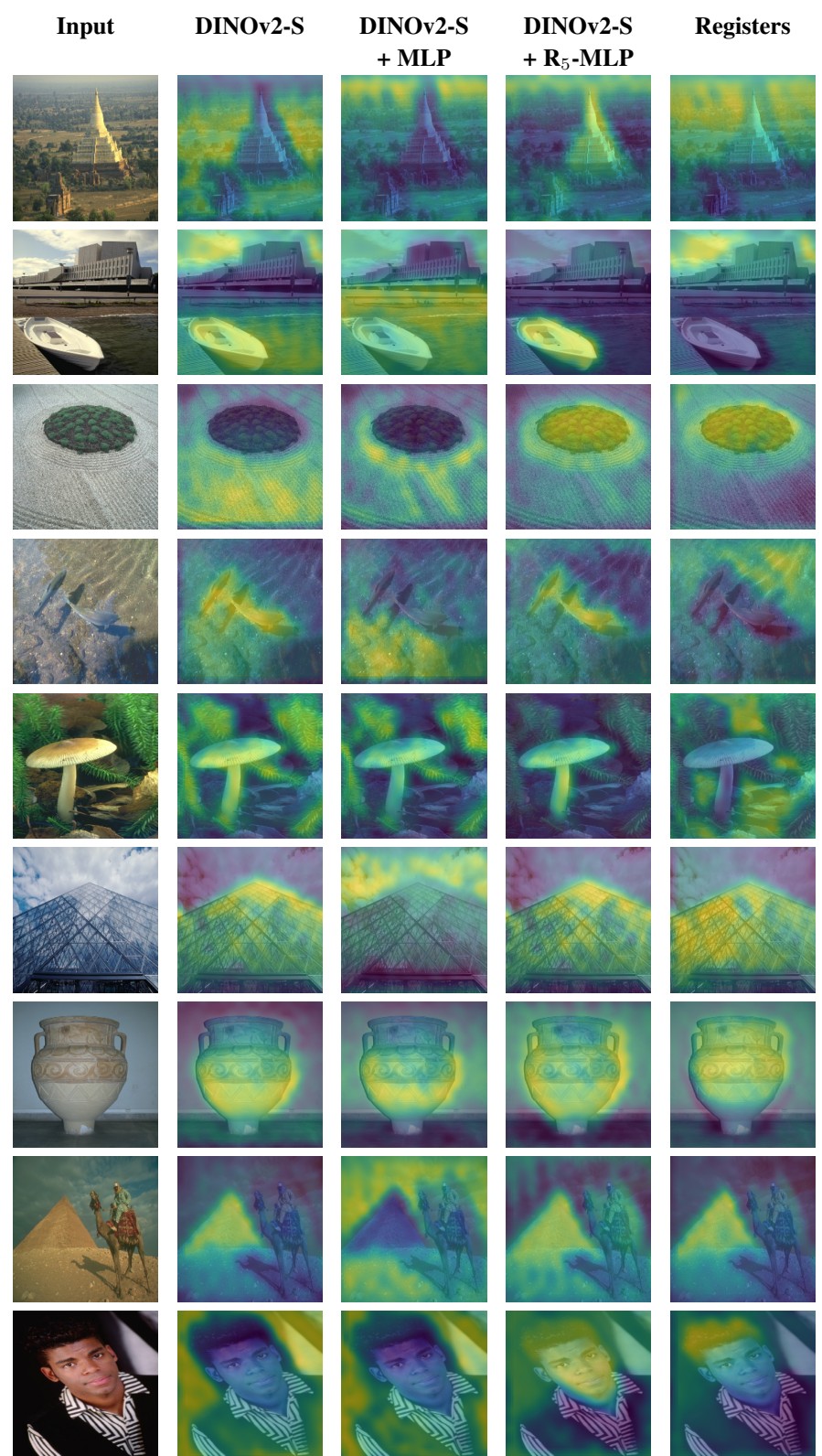

Figure 9: Second-order attention maps on the BSDS300 [27] dataset. Patch tokens were extracted using different backbones (columns).

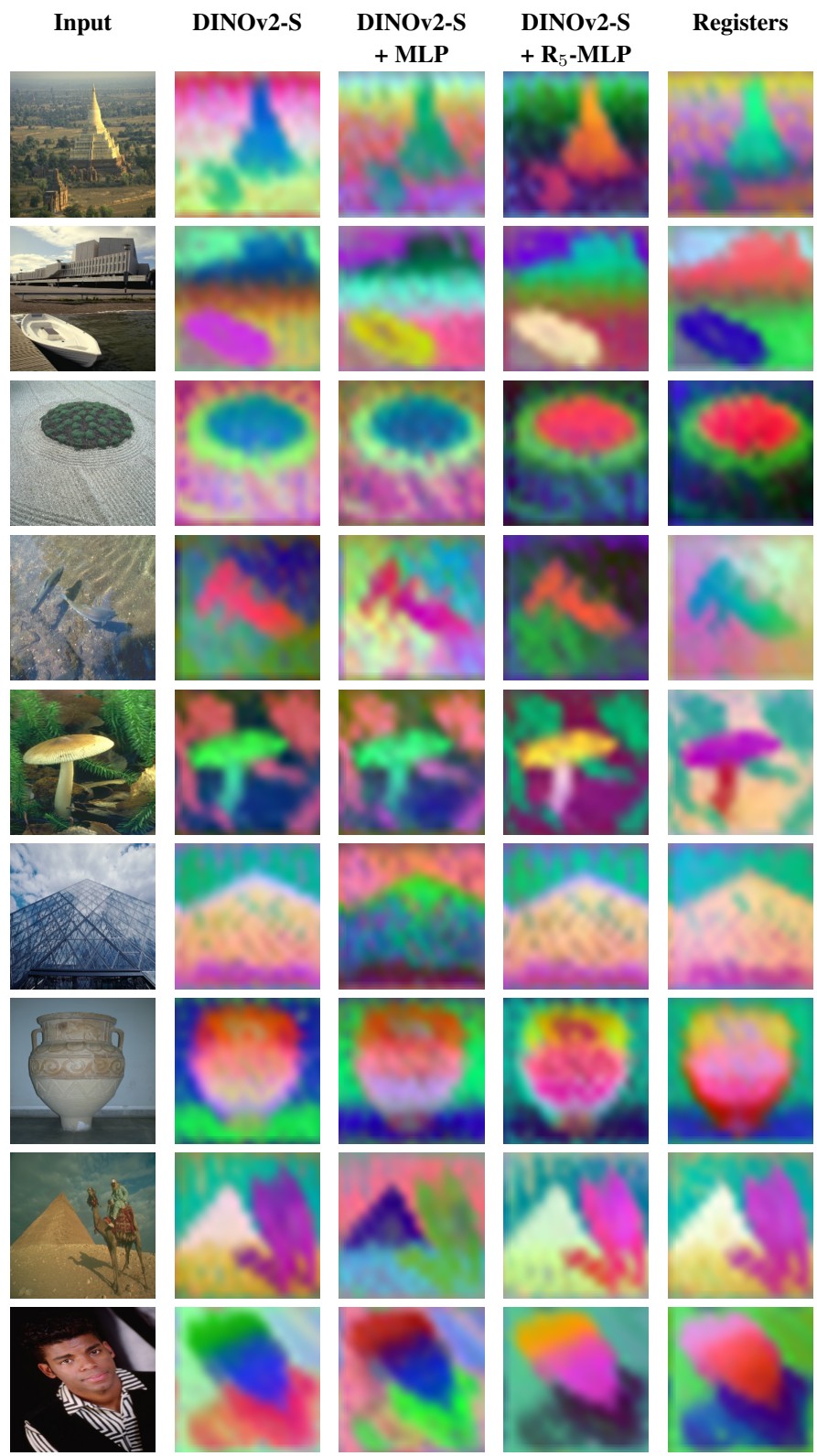

Figure 10: PCA visualization of embedding of BSDS300 [27] dataset using different backbones (columns).

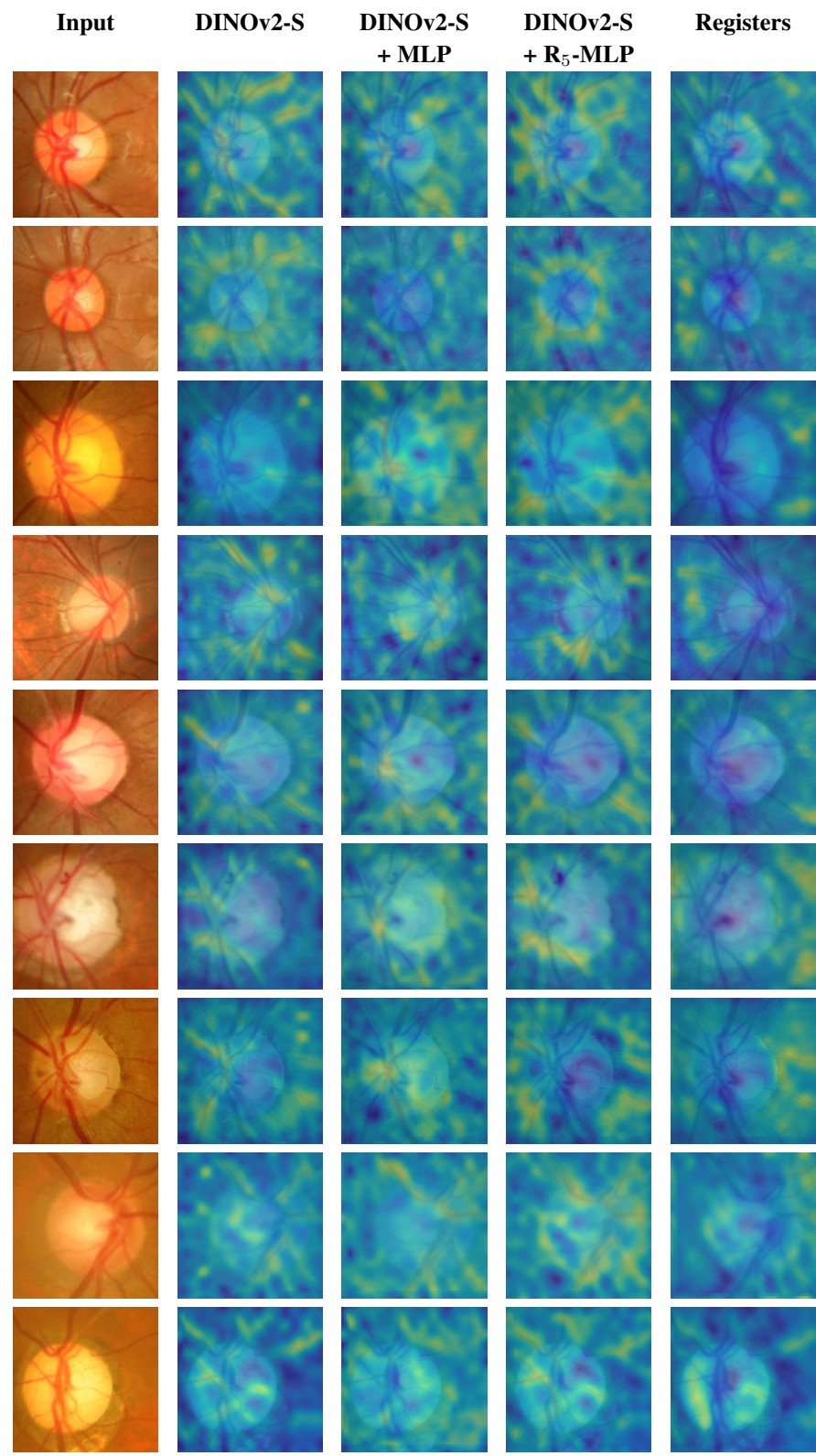

Figure 11: First-order attention maps on Glaucoma Fundus [2] dataset using different backbones (columns).

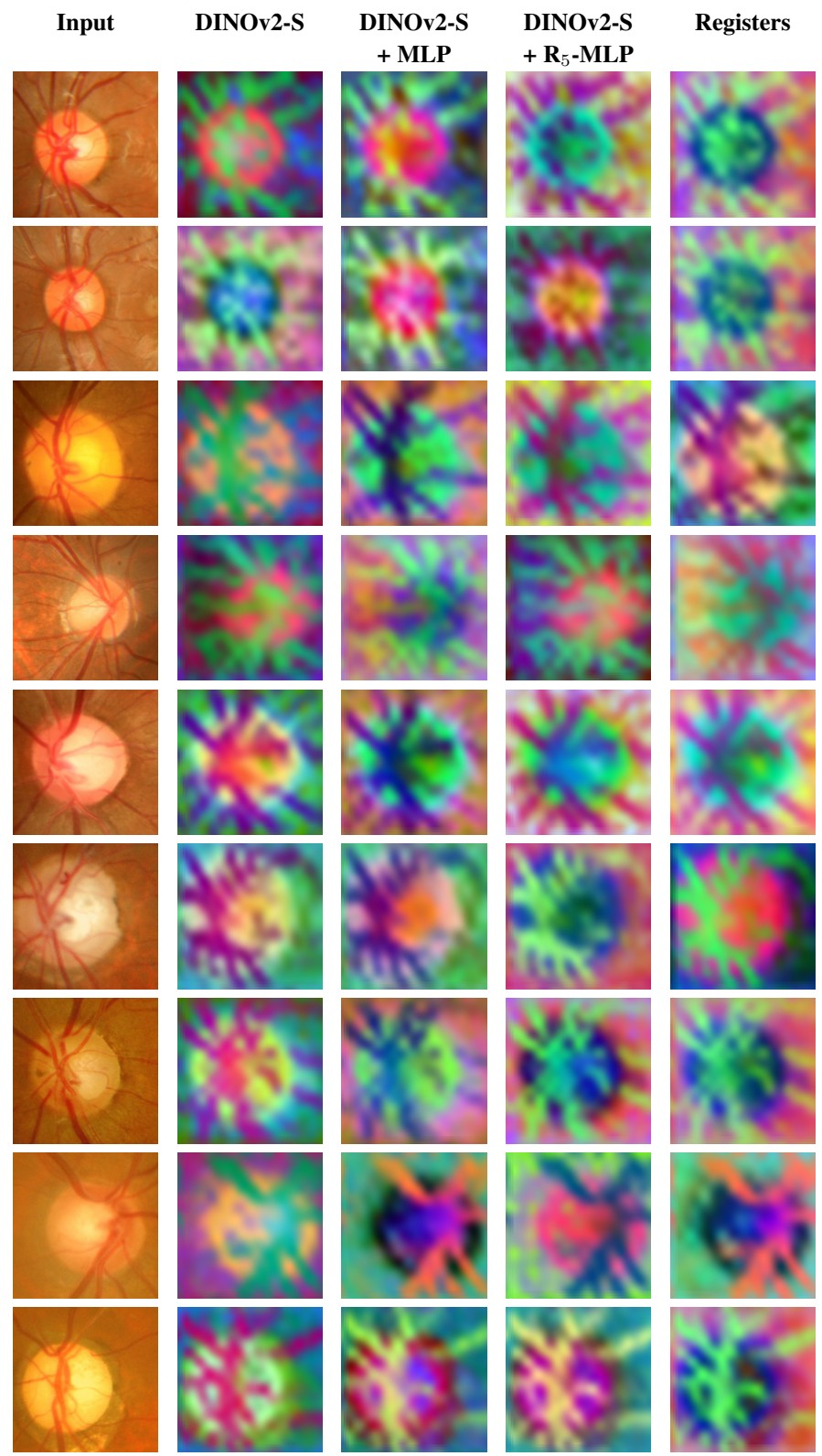

Figure 12: PCA visualization of embeddings of Glaucoma Fundus [2] dataset using different backbones (columns).

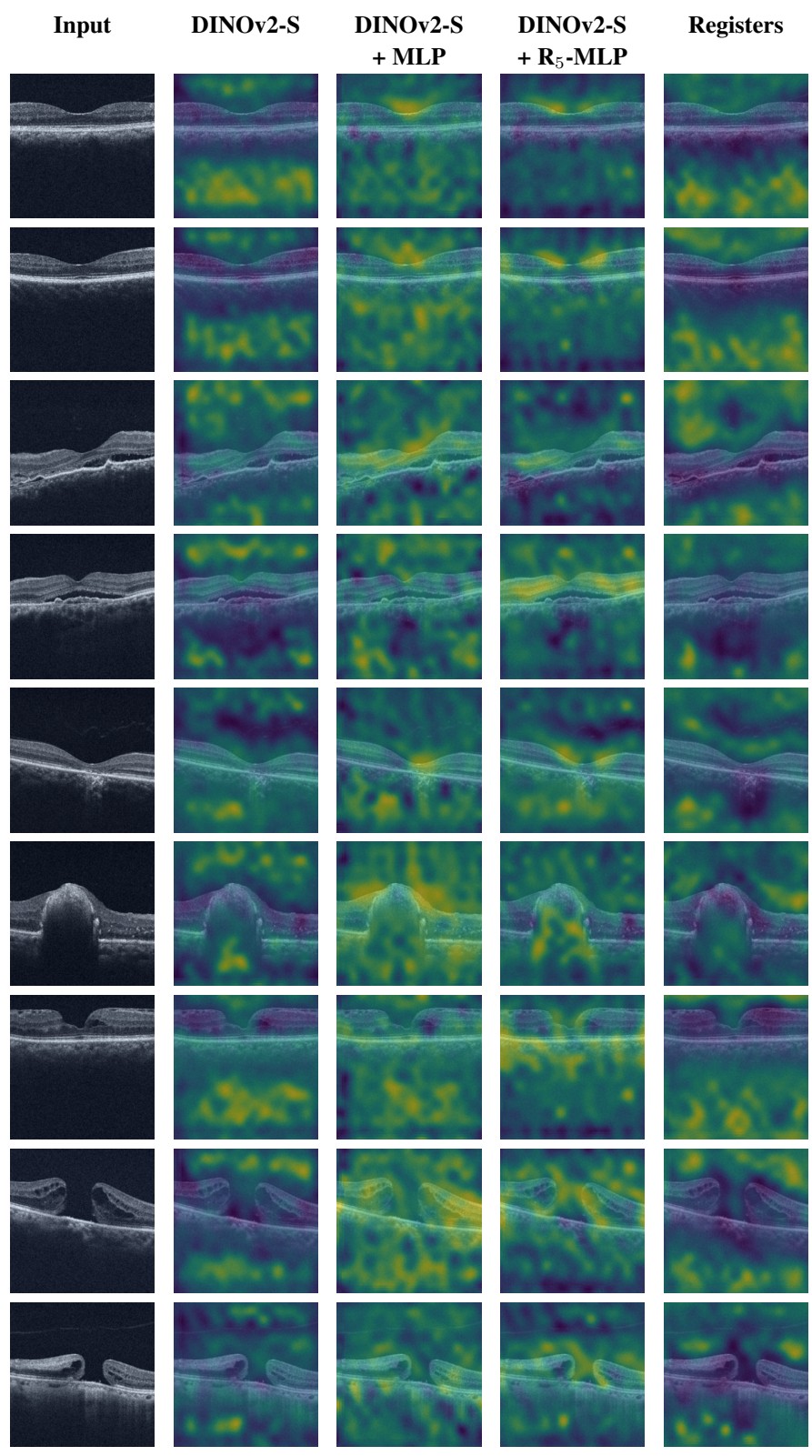

Figure 13: First-order attention maps on OCTID [14] dataset using different backbones (columns).

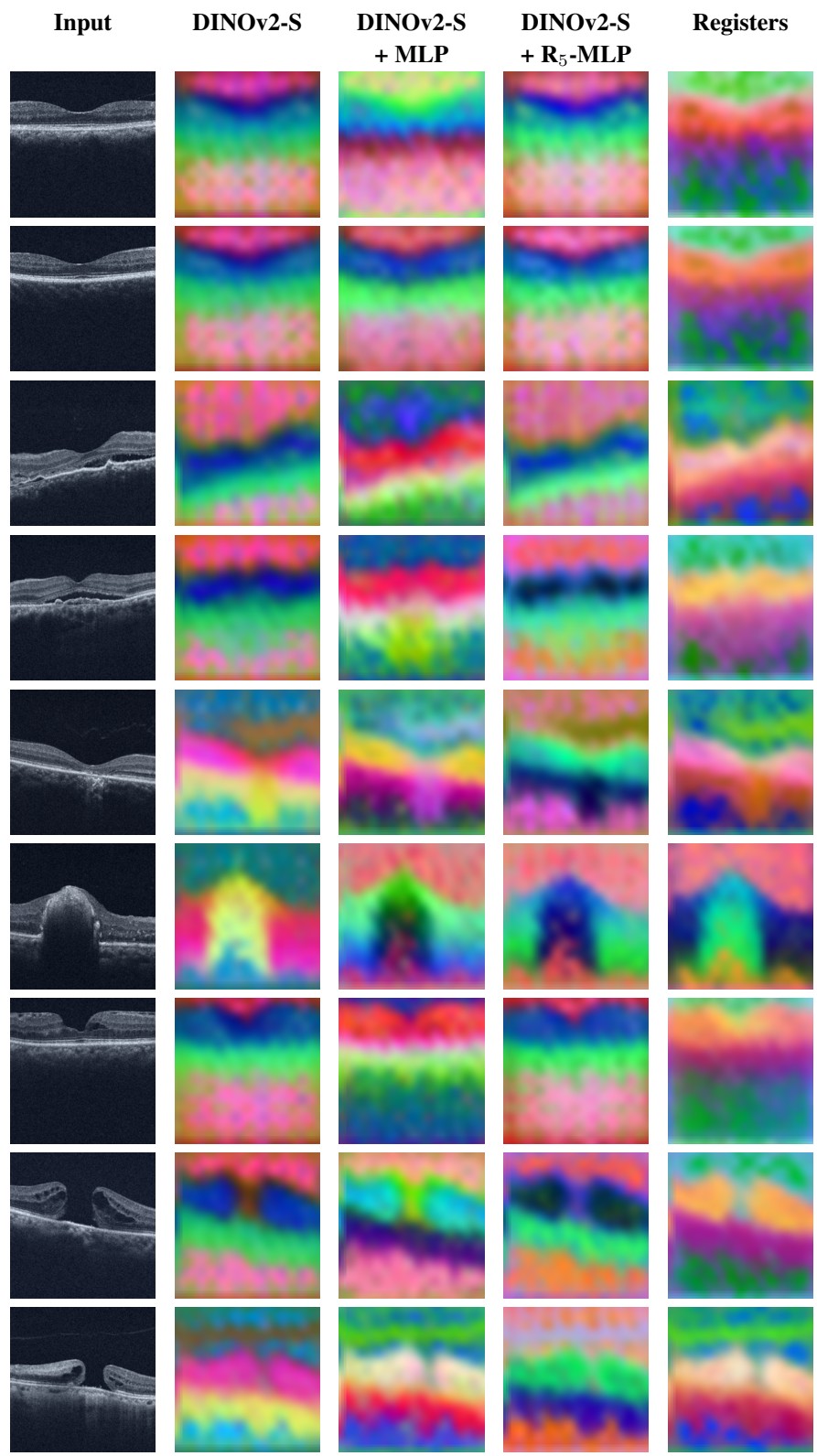

Figure 14: PCA visualization of embeddings of OCTID [14] dataset using different backbones (columns).

## A.2 Mathematical Definitions and Proofs

Given the interdisciplinary of the theoretical analysis in this paper, not only across scientific fields, but also across mathematical domains, we first provide a list of useful mathematical definitions results to analytically understand the results in this paper.

### A.2.1 General Mathematical Background

We present now a collection of topological definitions and a result.

**Definition 6.** *Given a function $f : X \to Y$, we will say that $f$ is locally injective on a subset $A \subseteq X$ if, $\forall x \in A$, $\exists U_x \subseteq X$ open such that $f|_{U_x}$ is injective.*

**Definition 7.** *Given a metric space $(X, \delta)$, a subset $A \subseteq X$ and $\varepsilon > 0$, we call a $\varepsilon$-cloud to the set $\{x \in X : \delta(x, A) < \varepsilon\}$.*

**Definition 8.** *Given a topological space $X$, we will say $X$ is disconnected if there are $A, B \subseteq X$ such that they are open and non-empty, $A \cap B = \varnothing$ and $A \cup B = X$. In addition, we will call $\{A, B\}$ a disconnection.*

**Definition 9.** *Given a topological space $X$ and $D \subseteq X$, we will say $D$ is dense in $X$ is for whatever open subset $U \subseteq X$, $U \cap D \neq \varnothing$*

**Remark 10.** *If $f : X \to Y$ is continuous and $Y$ is disconnected, $X$ is disconnected.*

*Proof.* Let $\{A, B\}$ be a disconnection for $Y$. Then $f^{-1}[A]$ and $f^{-1}[B]$ are open and non-empty due to $f$ being continuous and a function resp. Furthermore, $f^{-1}[A] \cup f^{-1}[B] = X$ since $f$ is a function and $\{A, B\}$ a disconnection. Lastly, using that $\{A, B\}$ is a disconnection and knowing that the pre-image opens finite intersections, $f^{-1}[A] \cap f^{-1}[B] = f^{-1}[A \cap B] = f^{-1}[\varnothing] = \varnothing$. □

### A.2.2 Topological Analysis of Vision Transformers

In this subsection, we will present a mathematical model describing ViTs and proving some results on their topological properties. Notably, the only assumption we are making in this subsection is that the corresponding ViT was trained using the KoLeo [35] regularizer.

**Theorem 11.** *ViTs can be decomposed as a tokenization function followed by a composition of translations. Said translations are defined by local orthonormal bases generated by the modulation of the data via the queries, keys and values layers.*

*Proof.* Let $\Omega$ be an image domain and $\Psi := \bigcup_{n \in \mathbb{N}} \mathbb{R}^{n \times d}$ be the ViT's latent space and $d$ its token dimension. Further, let $\mathcal{V} : \Omega \to \Psi$ denote the ViT as a learnable function and $T : \Psi \to \Psi$ the transformer encoder, following the structure from Dosovitskiy et al. [11]. We decompose these maps as $\mathcal{V} = T \circ \mathcal{C}$ and $T = \mathcal{N} \circ T_{L-1} \circ ... \circ T_0$, respectively where $\mathcal{C}$ is a continuous map from $\Omega$ to $\Psi$, $\mathcal{N}$ is a normalization layer and for each $l \in \{0, \ldots, L-1\}$, we define

$$T_l : \Psi \to \Psi$$
$$x \mapsto x + s_l(x)$$

with $s_l : \Psi \to \Psi$ described below.

Leaving the normalization $\mathcal{N}$ aside, we can see $T$ as a translation defined as a composition of the residual steps $s_l$. Following Vaswani et al. [39], $s_l$ can be further decomposed as

$$s_l : x \mapsto f_l \left( \mathcal{N}_{2,l}(x + h_l(\mathcal{N}_{1,l}(x))) \right)$$

where $h_l$ represents the multi-head self-attention mechanism, $f_l$ is the MLP block and $\mathcal{N}_{1,l}$ and $\mathcal{N}_{2,l}$ are layer normalizations.

Assuming linear layers have no bias for simplicity, the attention $h_l$ can be expressed as

$$h_l : x \mapsto \sigma \left( \frac{1}{r} \mathcal{L}_{Q,l}(xx^T)\mathcal{L}_{K,l}^T \right) \mathcal{L}_{V,l} x$$

where $\sigma$ is the softmax function, $r$ is a scaling factor and $\mathcal{L}_{V,l}, \mathcal{L}_{K,l}, \mathcal{L}_{Q,l}$ are learnable linear maps for queries, keys, and values.

Since $xx^T$ is symmetric, it can be diagonalized, allowing $h_l$ to be rewritten as

$$h_l : x \mapsto \sigma \left( \frac{1}{r} \mathcal{L}'_{Q,l} D_x \mathcal{L}'_{K,l} \right) \mathcal{L}_{V,l} x$$

where $D_x$ is diagonal and $\mathcal{L}'_Q$ and $\mathcal{L}'_K$ incorporate the change of basis.

Thus, we can understand $T$ as a map creating a field for the embeddings to follow, where the self-attention layers dynamically generate local orthonormal bases over $\Psi$ that the queries, keys and value matrices then modulate while the MLPs $f_l$ refine the resulting directions. $\qquad \square$

**Lemma 12.** *If a continuous function $f : X \to Y$ is locally injective on a compact set $K \subseteq X$ and the topology on $X$ is induced by a metric $\delta$, then $\exists \varepsilon > 0$ such that $f$ is injective on*

$$\{x \in X : \delta(x, K) < \varepsilon\}.$$

*Proof.* Let $U_\varepsilon := \{x \in X : \delta(x, K) < \varepsilon\}$ and let us work by contradiction, i.e. we will assume $\forall \varepsilon > 0, \exists \{x_\varepsilon, y_\varepsilon\} \subseteq U_\varepsilon$ such that $x_\varepsilon \neq y_\varepsilon$ and $f(x_\varepsilon) = f(y_\varepsilon)$. Given this, we can construct two sequences such that for $n \in \mathbb{N}^+$, $\{x_n, y_n\} \subseteq U_{1/n}$, $x_n \neq y_n$ and $f(x_n) = f(y_n)$.

Since $K$ is compact, it is easy to see that for every positive natural number $\overline{U_{1/n}}$ is compact too. Thus, given $X$ is metric and $\{x_n\}_{n>0}, \{y_n\}_{n>0} \subseteq \overline{U_1}$, we can extract convergent subsequences $\{x_{n_i}\}_{i>0}, \{y_{n_i}\}_{i>0}$ to $x$ and $y$ respectively.

Let us notice that $K = \bigcap_{n>0} U_{1/n}$. Therefore $\{x, y\} \subseteq K \Rightarrow f(x) = f(y)$ since $f$ is continuous and injective in $K$, having $x = y$ as a consequence. Also, since $f$ is locally injective, $\exists V_x$ open such that $x \in V_x$ and $f$ is injective on it and since $\{x_{n_i}\}_{i>0}, \{y_{n_i}\}_{i>0}$ both converge to $x$, $\exists N \in \mathbb{N}$ such that $\forall i > N, \{x_{n_i}, y_{n_i}\} \subset V_x \Rightarrow f(x_{n_i}) = f(y_{n_i}) \forall i > N$, reaching this way the contradiction we were looking for.

$\qquad \square$

**Lemma 13.** *If a function $f : X \to Y$ is injective and continuous with compact domain and Hausdorff codomain, $f$ is an homeomorphism on $f[X]$.*

*Proof.* Assuming the hypothesis from the lemma, the only thing remaining to show is that $f$ is open.

Thing being this way, let us take an open set $U \subseteq X$. Then $X \backslash U$ is closed which turns it into a compact set since $X$ is compact. Thus, by continuity, $f[X \backslash U]$ is compact, which makes it closed since $Y$ is Hausdorff, forcing $f[U]$ to be open.

$\qquad \square$

### A.2.3 RMLPs as Stochastic Regularizer for ViTs

We now introduce some useful definitions and rigorous proofs on the way RMLPs regularize the representation's topology by making the ViT perceive points in the representation space as ball of certain radius with high probability.

**Definition 14.** *[28] Let $E \subset \mathbb{F}^n$ a set and $\varepsilon \in (0, 1)$ be a distortion parameter. We say that a linear map $S : \mathbb{F}^n \to \mathbb{F}^m$ produces an $\varepsilon$-distortion of $E$ if, $\forall x \in E$,*

$$(1 - \varepsilon) \|x\|_2 \leqslant \|Sx\|_2 \leqslant (1 + \varepsilon) \|x\|_2.$$

**Remark 15.** *If $x \in \mathbb{R}^d$ and $\Gamma$ is a $n \times d$ matrix with iid entries and such that $\Gamma_{i,j} \sim \mathcal{N}(0, \sigma^2)$, then $\mathbb{E}[\|\Gamma x\|_2^2] = d\sigma^2 \|x\|_2^2$.*

*Proof.* Computing

$$\mathbb{E}[\|\Gamma x\|_2^2] = \mathbb{E}[\Sigma_i \, (\Sigma_j \Gamma_{ij} x_j)^2] = \Sigma_i \mathbb{E}[(\Sigma_j \Gamma_{ij} x_j)^2].$$

Since the entries of $\Gamma$ are independent and $\Gamma_{i,j} \sim \mathcal{N}(0, \sigma^2)$, we then have

$$\mathbb{E}[\|\Gamma x\|_2^2] = \Sigma_i \Sigma_j x_j^2 \mathbb{E}[\Gamma_{jj}^2] = d\sigma^2 \|x\|_2^2.$$

$\square$

**Definition 16.** *Let $S \in \mathbb{F}^{m \times n}$ be a linear map and $E \subseteq \mathbb{S}^{n-1}(\mathbb{F})$ a subset of the unit sphere in $\mathbb{F}^n$. Then, Martinsson and Tropp [28] define the minimum and maximum restricted singular values respectively as*

$$\sigma_{\min}(S, E) := \min_{x \in E} \|Sx\| \qquad and \qquad \sigma_{\max}(S, E) := \max_{x \in E} \|Sx\|.$$

**Lemma 17.** *[28] Let us consider $\{a_1, \ldots, a_N\} \subseteq \mathbb{R}^d$, $\Gamma \in \mathbb{R}^{n \times d}$ and build*

$$E = \left\{ \frac{a_i - a_j}{\|a_i - a_j\|} : 1 \leqslant i < j \leqslant N \right\} \subseteq \mathbb{S}^{d-1}.$$

*Then we have the following probability bounds:*

$$\mathbb{P}\left\{ \sigma_{\min}(\Gamma, E) \leqslant 1 - \frac{1 + 2\sqrt{\log(N/2)}}{\sqrt{n}} - t \right\} \leqslant e^{-dt^2/2}$$

*and*

$$\mathbb{P}\left\{ \sigma_{\max}(\Gamma, E) \geqslant 1 + \frac{2\sqrt{\log(N/2)}}{\sqrt{d}} + 1 \right\} \leqslant e^{-dt^2/2}.$$

*Thus, it is sufficient that $n \geqslant 8\varepsilon^{-2} \log N$ for being able to guarantee $\Gamma$ has a distortion of $\varepsilon$ 14 with high probability.*

**Corollary 18.** *If $\mathcal{A} \subseteq \Omega$ is the training data, and $T$ is locally injective on $\mathcal{A}$, the following holds:*

- ⊕) *There exists $\varepsilon > 0$ such that $T$ is an homeomorphism on an $\varepsilon$-cloud containing $\mathcal{A}$ (see Def. 7).*

- ·) *Let $P, Q \subseteq \Omega$. $\mathcal{C}[P] \cup \mathcal{C}[Q]$ is disconnected in $\Psi$ if and only if $\mathcal{V}[P] \cup \mathcal{V}[Q]$ is disconnected in $\Psi$ (see Def. 8), which can contribute to the batch effect.*

- :) *If $D \subseteq \Psi$ is dense in $\Psi$ (see Def. 9), then $\mathcal{V}[D]$ is dense in $\mathcal{V}[\Psi]$.*

*Proof.* Assuming the corollary hypothesis, let us prove the statements.

⊕) The existence of the $\varepsilon$-cloud is direct result of theorem 2.

·) The result follows directly from Theorem 2 since homeomorphisms preserve connections and disconnections.

:) Let $D$ be dense in $\Psi$. We note $\mathcal{C}$ is continuous for being defined as a multiplication of matrices and addition of vectors. Thus $\mathcal{C}[D]$ is dense in $\mathcal{C}[\Omega]$. Therefore, $\mathcal{V}[D]$ is dense in $\mathcal{V}[\Omega]$ since $T$ is an homeomorphism because of theorem 2.

$\square$

**Theorem 19.** *Being $\{p_1, \ldots, p_N\} \subseteq R^m$, $\varepsilon > 0$, $\lambda > 0$ and $\Gamma$ a matrix of size $n \times m$ whose entries are iid and following a normal distribution $\mathcal{N}(0, \lambda n^{-1})$, $\Gamma$ will have an $\varepsilon$ distortion on the set*

$$E := \left\{ \frac{p_i - p_j}{\|p_i - p_j\|} : 1 \leqslant i < j \leqslant N \right\}$$

*with high probability if*

$$\lambda n^{-1} < \frac{\varepsilon^2}{8 \ln N}.$$

*In addition,*

$$\mathbb{E}[\|\Gamma x\|_2^2] = m\lambda n^{-1}\|x\|_2^2.$$

*Proof.* To see $\mathbb{E}[\|\Gamma x\|_2^2] = m\lambda n^{-1}\|x\|_2^2$, it is enough to use remark 15.

On the other hand, to compute the distortion of $\Gamma$, let us assume the hypothesis and define the restricted singular values following Martinsson and Tropp [28] as

$$\sigma_{\min}(\Gamma, E) := \min_{x \in E} \|\Gamma x\| \qquad \sigma_{\max}(\Gamma, E) := \max_{x \in E} \|\Gamma x\|,$$

let us realize the statement is equivalent to having

$$1 - \varepsilon < \sigma_{\min}(\Gamma, E) \leqslant \|\Gamma x\| \leqslant \sigma_{max}(\Gamma, E) < 1 + \varepsilon.$$

Thus, having $\sqrt{\lambda/d} < \varepsilon/\sqrt{8 \ln N}$, it follows $\varepsilon > \sqrt{\lambda/n}\left(1 + 2\sqrt{\ln(N/2)}\right)$ and then $1 - \varepsilon < 1 - \sqrt{\lambda/n}\left(1 + 2\sqrt{\ln(N/2)}\right)$. This way, we can take $t > 0$ and write

$$\mathbb{P}\left(\sigma_{\min}(\Gamma, E) \leqslant 1 - \varepsilon - t\right) \leqslant \mathbb{P}\left(\sigma_{\min}(\Gamma, E) \leqslant 1 - \sqrt{\lambda/n}\left(1 + 2\sqrt{\ln(N/2)}\right) - t\right).$$

Similarly, $\sqrt{\lambda/n} < \varepsilon/\sqrt{8 \ln N}$ implies $1 + \varepsilon > 1 + 2\sqrt{\lambda n^{-1} \ln(N/2)}$. Thus, being $t > 0$,

$$\mathbb{P}\left(\sigma_{\max}(\Gamma, E) \geqslant 1 + \varepsilon + t\right) \leqslant \mathbb{P}\left(\sigma_{\max}(\Gamma, E) \geqslant 1 + 2\sqrt{\lambda n^{-1} \ln(N/2)} + t\right).$$

Therefore, using that the Gaussian width of $E$, $w(E)$, satisfies $w(E) < 2\ln(N/2)$ and the theorem for restricted singular values and a Gaussian matrix from Martinsson and Tropp [28], for every $t > 0$ we have

$$\mathbb{P}\left(\sigma_{\min}(\Gamma, E) \leqslant 1 - \varepsilon - t\right) \leqslant e^{-\lambda^{-1}nt^2/2}$$

and

$$\mathbb{P}\left(\sigma_{\max}(\Gamma, E) \geqslant 1 + \varepsilon + t\right) \leqslant e^{-\lambda^{-1}nt^2/2}.$$

$\therefore$ $\Gamma$ has an $\varepsilon$-distortion with high probability. $\qquad \square$

## A.3 Technical Details

**Backbone Training.** All training was performed on a single GPU (Quadro RTX153 8000 or NVIDIA A100-SXM4-40GB) and required approximately 15 hours per trained backbone, reflecting the low computational cost of our approach and its reduced environmental footprint. Models were trained on randomly sampled mini-batches until validation performance plateaued. For datasets without predefined splits, we manually partitioned them into 70% training, 15% validation, and 15% testing.

All models were trained using the AdamW optimizer [24], which we employed consistently across fine-tuning stages as well as during training of downstream linear heads and UNet decoders.

Table 5 shows the main hyperparameters used when fine-tuning DINOv2-S on natural, OCT and CFP modalities. Further implementation details can be found in our code.

**Downstream Tasks Training.** For classification tasks using DINOv2, we constructed the input representations for 1-Nearest Neighbor, Random Forest, and linear probing classifiers by concatenating the class token with the mean of the patch tokens. In contrast, for SwAV, which does not produce patch tokens, we used its output directly. The linear classifier was trained using the cross-entropy loss.

Table 5: Hyperparameters used for fine-tuning DINOv2-S to obtain C-ViT and $R_\lambda$-ViT.

| Hyperparameter | Value |
|---|---|
| Optimizer | AdamW[24] |
| Plateau size for early stop | 10 epochs |
| Batch size | 32 |
| Token's dimension | 384 |
| DINO coefficient | 1 |
| iBOT coefficient | 1 |
| KoLeo coefficient | 0.5 |
| Initial learning rate | 1e-7 |
| Patience/factor for learning rate scheduler | 3/0.4 |
| Minimum learning rate | 1e-8 |
| Hidden/Bottleneck/Output dimensions for MLPs and RMLPs | 1536/256/65536 |
| Number of transformer blocks | 12 |
| Patch size | 14 |
| Crop size | 224 |
| Steps per epoch | 100 |
| Warm up epochs | 10 |

To integrate the ViT and UNet architectures for dense prediction tasks, we first projected the output of the ViT through a linear layer and then concatenated it with the features from the encoder branch of the UNet. This combined representation was subsequently fed into the UNet's decoder branch. The ViT outputs were handled differently depending on the task: for segmentation, we concatenated the class token with the patch tokens, while for depth estimation, only the patch tokens were used. Segmentation heads were trained using a weighted combination of focal loss and Dice loss, whereas depth estimation heads were trained using focal loss.

**Evaluation of Patch Token Quality**. First-order attention maps were computed by evaluating the norm of the patch token embeddings. For the second-order attention maps, we independently performed principal component analysis on the patch tokens of each image and calculated the norm of the top three principal components. To distinguish between low- and high-information patches within an image, we first computed the image gradient, applied a smoothing operation, and then averaged the resulting gradient values within each patch. A Gaussian Mixture Model with two components was subsequently used to classify the patches based on their average gradient magnitudes.

## A.4 Information on External Sources

We used two pre-trained models in this work. Their licenses and repositories can be found in Table 6. A recollection of all the datasets used in this work can be found in Table 7a, both for natural and medical domains. To show the geographic diversity from our geographical dataset, we show the country of origin of the medical datasets used in this paper in Table 7b.

Table 6: External code and weights used as baselines.

| Name | License | Repository |
|---|---|---|
| DINOv2 [30] | Apache-2.0 | https://github.com/facebookresearch/dinov2 |
| RETFound [43] | CC BY-NC 4.0 | https://github.com/rmaphoh/RETFound_MAE |
| DVT [42] | MIT License | https://github.com/Jiawei-Yang/Denoising-ViT |
| Registers [9] | Apache-2.0 | https://github.com/facebookresearch/dinov2 |
| SwAV [4] | License: CC BY-NC 4.0 | https://github.com/facebookresearch/swav |
| Sinder [40] | - | https://github.com/haoqiwang/sinder |

Table 7: Used datasets, licenses and country of origin.

(a) Used datasets and their licenses.

| Name | License | Repository |
|---|---|---|
| ImageNet-1k [34] | CC0: Public Domain | ImageNet-1k |
| NYU-Depth V2 [36] | MIT | NYU-Depth V2 |
| ADE20k [45] | BSD-3-Clause | ADE20k |
| BSDS300 [27] | Non-commercial research | BDSD300 |
| VOC07 [13] | Custom | PASCAL VOC 2007 |
| OCTID [14] | CC0 1.0 | OCTID |
| Glaucoma Fundus [2] | CC0 1.0 | Glaucoma Fundus |
| IDRID [31] | Open Access | IDRID |
| JSIEC [7] | Open Access | JSIEC |
| MESSIDOR-2 [1, 10] | Non-commercial research | MESSIDOR-2 |
| PAPILA [22] | GPL 3.0+ | PAPILA |
| Retina [6] | Open Access | Retina |
| Aptos [3] | Custom | Aptos |
| Eckardt, et al. [12] | Property of LMU University Hospital | - |

(b) Country of creation for medical dataset.

| Dataset | Country |
|---|---|
| OCTID [14] | India |
| Glaucoma Fundus [2] | South Korea |
| IDRID [31] | India |
| JSIEC [7] | China |
| MESSIDOR-2 [1, 10] | France |
| PAPILA [22] | Spain |
| Retina [6] | Unspecified |
| Eckardt, et al. [12] | Germany |

## A.5 Supplementary Results

Table 8: DICE scores for segmentation on the Eckardt, et al. [12] dataset with emphasis on the Outer Nuclear Layer (ONL).

| Model | ViT-UNet hybrid | | Linear Head | |
|---|---|---|---|---|
| | Averaged DICE | DICE on ONL | Averaged DICE | DICE on ONL |
| RETFound [43] | $0.92\pm0.06$ | $0.59\pm0.10$ | $0.10\pm0.02$ | $0.01\pm0.02$ |
| DINOv2-S [30] | $0.79\pm0.20$ | $0.54\pm0.20$ | $^{\dagger}0.16\pm0.08$ | $\mathbf{0.12\pm0.05}^{***}$ |
| DINOv2-S+MLP | $0.86\pm0.20$ | $0.55\pm0.20$ | $\underline{\mathbf{0.20\pm0.05}}$ | $0.03\pm0.03$ |
| DINOv2-S+$R_{0.1}$-MLP | $^{\dagger}0.94\pm0.12$ | $^{\dagger}0.68\pm0.21$ | $\underline{\mathbf{0.20\pm0.03}}$ | $0.06\pm0.04$ |
| DINOv2-S+$R_5$-MLP | $\mathbf{0.97\pm0.03}^{*}$ | $\mathbf{0.72\pm0.09}^{*}$ | $0.13\pm0.01$ | $^{\dagger}0.08\pm0.03^{**}$ |
| DINOv2-S+$R_{10}$-MLP | $0.87\pm0.20$ | $0.61\pm0.20$ | $\underline{\mathbf{0.20\pm0.01}}$ | $\underline{0.07\pm0.05}^{*}$ |
| DINOv2-S+$R_{20}$-MLP | $0.70\pm0.30$ | $0.47\pm0.20$ | $0.13\pm0.02$ | $\underline{0.07\pm0.05}^{*}$ |
| Eckardt, et al. [12] | $0.92\pm0.03$ | $0.44\pm0.03$ | – | – |

