# OpenReview forum: "Randomized-MLP Regularization Improves Domain Adaptation and Interpretability in DINOv2"
_NeurIPS.cc/2025/Conference — NeurIPS 2025 poster_

### Official Review · Reviewer_WhNa · 2025-07-03

**Clarity:** 3
**Significance:** 2
**Originality:** 3
**Rating:** 5
**Confidence:** 3

**Summary:**

This paper introduces a regularization technique called Randomized-Multi-Layer Perceptron (Randomized-MLP). Randomized MLP is a contrastive learning-based method that
This work conducts the experiment on top of DINOv2, and compares Randomized MLP trained model versus regular MLP fine tuned model, and shows Randomized MLP improves downstream tasks including semantic segmentation and depth estimation. Randomized MLP fine-tuned also show cleaner attention maps among nature and medical images. Authors conduct theoretical analysis to prove Randomized MLP impacts the contrastive learning paradigm by turning point embeddings into probability balls with the help of KoLeo regularizer.

**Questions:**

* In contrast with previous work, our results show that this behavior also emerges in smaller versions of the model => Do you mean storing global information in patch tokens? Do you mean DINOv2-S is worse?
* Line 87 “sparcity” => I believe you mean sparsity.

**Ethical Concerns:**

["NO or VERY MINOR ethics concerns only"]

**Final Justification:**

This work proposes a regularization method that improve downstream tasks and make the attention map more meaningful. Authors added results to generalize this method to non DINO models and also provided quantitive analysis on attention map alignment, which make the evaluation more comprehensive, thus I think the technical contribution is solid.

**Limitations:**

yes

**Quality:**

3

**Strengths And Weaknesses:**

Strength
* The methodology is intuitive and mathematical details are well explained. Authors explain the impact of each component, e.g., contrastive losses encourage similarity between views, KoLeo promotes uniformity, and GELU restricts the space to mostly positive coordinates. This paper also provides detailed visualization to explain the impact of different values of \lambda.
* The proposed method is not bound to a specific architecture and in theory it can be generalized to other domains, although the actual movement needs to be evaluated.
* Theoretical foundation: authors provide theoretical analysis on how the embedding is projected and how it provides regularization effect.
* Straightforward and comprehensive experiment setup. This paper evaluated Randomized MLP across different downstream tasks on different datasets.

Weakness
* It would be a plus if authors can show how this regularization works with other models, since it looks very generalizable and it should be easy to apply to other models. Does it only help DINO style models, or vision models, or can be generalized to NLP tasks?
* Although authors present the negatively skewed proportion curve reflecting the sparse representation, I don’t find quantitative results on to what extent the attention maps become cleaner or better aligned with human perception.

---

> ### Author Rebuttal · Authors · 2025-07-29
>
> Dear reviewer,
>
> We are glad to read that you found our experiment setup straightforward, our methodology intuitive and our mathematical results properly explained. We also want to thank you for your constructive feedback and would like to address your comments and questions in the following sections.
>
> **DINOv2-S artifacts on ophthalmological data**. We appreciate the opportunity to clarify this point. In stating that this behavior emerges in a smaller version of the model, we were referring to our observation that DINOv2-S places uninterpretable attention on void or low-informative regions in ophthalmological images (as shown in Figure 5). This phenomenon has been linked by Darcet et al. (2024) to global information being stored in patch tokens, but their analysis primarily noticed it on larger models. Our results indicate that this artifact also appears in DINOv2-S, suggesting that the issue is not limited to high-capacity models. We agree that this could be stated more clearly and will update the manuscript accordingly to avoid confusion.
>
> **Typo**. Good catch! Thank you. We will correct it.
>
> **RMLP implementation on other architectures**. Our theoretical analysis suggests that RMLPs can be applied broadly to any contrastive learning setup, independent of the underlying ViT architecture. To support this, we conducted some preliminary tests using the DINOv2 base model and observed a similar performance improvement by including RMLPs. Nevertheless, we decided not to include these results in this manuscript’s final version because we wanted to focus on the low-resource setup, to improve computational efficiency, a common challenge in the medical imaging field. Exciting follow-up projects could include investigating the effects of RMLPs not only on architectures like MoCo and Swin Transformers used in vision tasks, but also on those used for natural language processing, such as SimSCE or BERT.
>
> **New quantitative results (Correlating high-norm patches and anatomical structures)**. We acknowledge the importance of supporting qualitative observations with quantitative evidence and therefore conducted additional experiments to strengthen our interpretability claims. Using the dataset from Eckardt et al. (2024)—which contains OCT scans with expert-annotated retinal layers for patients with and without Inherited Retinal Disorders (IRDs)—we assessed the alignment between patch token norms and retinal structures. Punctually, we calculated the correlation between the spatial positions of the top 25% highest-norm patch tokens and the annotated retinal layer regions. The results show that DINOv2-S exhibits little to no alignment with anatomical structures, while R$_5$-ViT ($\lambda=5$) demonstrates a stronger correspondence, indicating improved localization of clinically meaningful features. These results offer quantitative backing for our interpretability claims, particularly within ophthalmology, and we will happily include them in the final version of our manuscript.
>
> |    Model    | Patients with IRD | Patients without IRD |
> |:-----------:|:-----------------:|:--------------------:|
> |  DINOv2-S   |       -0.06       |         0.09         |
> | R$_5$-ViT   |        0.36       |         0.33         |
>
> **New quantitative results (Information in patch tokens)**. We further examined model behavior by assessing the diagnostic relevance of individual patch tokens. Using a protocol similar to Darcet et al. (2024), we conducted a 1-nearest neighbor classification task on the Eckardt et al. (2024) dataset to distinguish patients with and without IRDs. Explicitly, we compared classification accuracy using: (1) the class token, (2) the highest-norm patch token, and (3) the lowest-norm patch token.
>
> As shown in the table below, DINOv2-S achieves decent accuracy with the class token (0.76), but its highest-norm patch token performs poorly (0.48)— worse than the lowest-norm token (0.76). This indicates that high-norm tokens may correspond to irrelevant or void regions. In contrast, R₅-ViT performs well with the class token (0.88) and retains high accuracy using the highest-norm patch token (0.76), while the lowest-norm token performs poorly (0.40), as expected. These results support our claim that RMLP promotes attention to diagnostically relevant regions, rather than repurposing low-information areas, and we will gladly include them in the final version of our paper.
>
> |   Model   | Class Token | Highest-Norm Patch Token | Lowest-Norm Patch Token |
> |:-------------:|:---------------:|:-----------------------------:|:----------------------------:|
> |   DINOv2-S    |      0.76       |             0.48             |            0.76             |
> |  R$_5$-ViT    |      0.88       |             0.76             |            0.40             |
>
>
> **References**
>
> Darcet, T., Oquab, M., Mairal, J., Bojanowski, P.: Vision transformers need registers. arXiv. arXiv:2309.16588 (2024)
>
> Eckardt, F., Mittas, R., et al.: Deep Learning-Based Retinal Layer Segmentation in Optical Coherence Tomography Scans of Patients with Inherited Retinal Diseases. Klin Monbl Augenheilkd. doi: 10.1055/a-2227-3742 (2024)

---

> > ### Comment · Reviewer_WhNa · 2025-08-05
> >
> > It is a big pity that authors didn't add results on non DINOv2 models. For a method that should be easily generalizable, missing those results significantly decreases the significance of this work.
> >
> > Thanks for adding new quantitative results.  "Punctually, we calculated the correlation between the spatial positions of the top 25% highest-norm patch tokens and the annotated retinal layer regions."  => Can you explain the process with more details? Did you do a 1 to 1 match of the top 25% tokens? Or just calculate the overlap?

---

> > > ### Author Response · Authors · 2025-08-07
> > >
> > > Dear Reviewer,
> > >
> > > We are glad you appreciated the new quantitative results we added. We would like to address your new questions on the following sections.
> > >
> > > **Experiments on non-DINOv2 models**. We fully agree that a key strength of our method lies in its generalizability and ease of implementation across different model architectures. In light of this, and acknowledging that including experiments fine-tuning non-DINOv2 models would enhance the impact of our work, we conducted preliminary experiments fine-tuning the MoCo v3 model [c] on the OCT dataset from [a]. For these experiments, we employed a classical MLP head as well as our proposed RMLP regularizer with $\lambda=5$, following the training procedure described in the paper.
> > > After training the backbone, we evaluated the model on downstream tasks similar to those presented in our manuscript. Specifically, we performed classification experiments on the dataset from [a] and semantic segmentation experiments on the dataset from [b]. The results of these additional experiments are summarized in the tables below, alongside the relevant original results from our submission. Bold models are the added ones.
> > >
> > > We emphasize that these results are preliminary due to time constraints during the discussion period. Nonetheless, we find their promising nature remarkable, as they demonstrate improved downstream task performance when using the RMLP head compared to a classical MLP. This behavior aligns with our observations on DINOv2-S. Additionally, it is worth highlighting that MoCo v3 employs a ViT backbone with a larger latent dimension than DINOv2-S, which further suggests that our regularization method effectively generalizes across different latent spaces.
> > >
> > > We sincerely thank the reviewer for the suggestion to conduct these experiments. We intend to include them in the final version of the paper, accompanied by a robust statistical analysis comparable to what we provided for the models originally presented.
> > >
> > > **Semantic segmentation results**
> > >
> > > | Model                       | Mean DICE ViT-UNet hybrid | DICE on ONL ViT-UNet hybrid |
> > > |:---------------------------:|:-------------------------:|:---------------------------:|
> > > | **MoCov3+MLP**              | 0.97                      | 0.69                        |
> > > | **MoCov3+RMLP ($\lambda=5$)** | 0.98                      | 0.73                        |
> > > | RetFound                    | 0.92 ± 0.06               | 0.59 ± 0.1                  |
> > > | DINOv2-S                   | 0.79 ± 0.2                | 0.54 ± 0.2                  |
> > > | C-ViT                       | 0.86 ± 0.2                | 0.55 ± 0.2                  |
> > > | R$_5$-ViT                  | 0.97 ± 0.03               | 0.72 ± 0.09                 |
> > >
> > >
> > > **Classification results**
> > >
> > > | Model                        | 1-NN        | Random Forest | Linear Classification   |
> > > |:----------------------------:|:-----------:|:-------------:|:-----------------------:|
> > > | **MoCov3 + MLP**             | 0.76        | 0.64          | 0.55                    |
> > > | **MoCov3 + RMLP ($\lambda=5$)** | 0.76        | 0.66          | 0.59                    |
> > > | RetFound                    | 0.8         | 0.81          | 0.93 ± 0.05             |
> > > | DINOv2-S                   | 0.75        | 0.75          | 0.88 ± 0.08             |
> > > | C-ViT                       | 0.59 ± 0.02 | 0.51 ± 0.01   | 0.76 ± 0.02             |
> > > | R$_5$-ViT                  | 0.78 ± 0.01 | 0.74 ± 0.01   | 0.87 ± 0.01             |
> > >
> > >
> > > **Correlation analysis between the highest-norm patch tokens and annotated retinal layers** For each OCT slice, we extracted patch tokens and ranked them by their norm. We then generated a binary mask identifying patches in the top 25% norm as 1, with the rest marked as 0. Using domain-expert annotations, we created a corresponding binary mask indicating the presence of retinal layers per patch. We computed the Pearson correlation coefficient between these two binary masks on a per-OCT scan basis and reported the average value across the whole dataset. This method statistically captures the spatial overlap between high-norm patches and retinal layers, rather than enforcing a strict one-to-one token matching.
> > >
> > > **References**
> > >
> > > [a] Gholami, P. et al.: OCTID: optical coherence tomography image database. Comput. Electr. Eng. 81, 106532 (2020)
> > >
> > > [b] Eckardt, F., Mittas, R., et al.: Deep Learning-Based Retinal Layer Segmentation in Optical Coherence Tomography Scans of Patients with Inherited Retinal Diseases. Klin Monbl Augenheilkd. doi: 10.1055/a-2227-3742 (2024)
> > >
> > > [c] Chen, Xinlei, et al. Improved baselines with momentum contrastive learning arXiv preprint arXiv:2104.02057 (2021).

---

> > > > ### Comment · Reviewer_WhNa · 2025-08-08
> > > >
> > > > Thanks for the additional results. Although they are preliminary, I think the addition evaluation make this work more convincing and solid, and addressed most of my concerns.

---

### Official Review · Reviewer_UqKd · 2025-07-03

**Clarity:** 2
**Significance:** 2
**Originality:** 3
**Rating:** 4
**Confidence:** 4

**Summary:**

This paper aims to improve the interpretability of the attention maps and feature maps of DINO ViTs, especially for medical images. They propose to replace the MLP heads of DINO with randomized MLPs that are fixed and add Gaussian noise to the intermediate features. They prove that adding Gaussian noise parameterizes the features to probability balls, and KoLeo promotes feature uniformity. They show that the scale of the noise should not be too large (Fig. 2). In experiments, the method shows better attention maps and improved performance on some benchmarks.

**Questions:**

Can the authors provide more details on how DINO is re-implemented, and why weight normalization and feature normalization are omitted?

**Ethical Concerns:**

["NO or VERY MINOR ethics concerns only"]

**Final Justification:**

The discussion has resolved most of my concerns and the remaining ones can be resolved in a revision.

**Limitations:**

The limitations are discussed on page 9.

**Paper Formatting Concerns:**

No significant formatting issues. The checklist is printed twice though.

**Quality:**

3

**Strengths And Weaknesses:**

### Strengths
- The paper considers improving the interpretability of pre-trained DINO features for medical usage, which is beneficial for social good.
- The paper provides mathematical proof for some properties of DINO and the proposed RMLP.
- The paper also provides visualizations to depict the properties of DINO and the proposed method (eg, Fig. 6).

### Weaknesses
- I noticed that in the provided code, the implementation of both DINOHead and the modified version omitted weight normalization (on the last layer) and also feature normalization. These are crucial designs of DINO to ensure that self-distillation is performed on normalized *prototypes*, and removing them has shown notable performance drop in the original paper. This could raise a critical concern about whether DINO is faithfully re-implemented and whether the comparison is fair. Why did the authors remove these operations?
- The comparisons, especially visualizations of attention maps and PCA visualizations, are always provided in very low resolutions. In DINO, DINOv2, and Darcet et al., they provided higher-resolution visualizations that are much clearer for comparison between methods. It is strongly suggested that the authors also update these visualizations to higher-res ones, as currently it is hard to identify important parts, e.g., vessels in medical images.
- Many conclusions in the proof (Sec. 4) are unsurprising and this part could be moved to later sections for clarity. For example, KoLeo regularizer is not an important part of DINOv2, and removing it does not affect model behaviour notably. The main regularizers are the centering and sharpening operations, and I wonder why KoLeo is picked for analysis. Also, the fact that adding Gaussian noise turns deterministic inputs into stochastic "probability balls", and the scale of the noise should be small, are trivial.
- The structure and formatting could also be improved. Currently, the introduction section is overly simple, and the related work section is also missing. Besides, I would also suggest adding a formal introduction of DINO in the main text.

---

> ### Author Rebuttal · Authors · 2025-07-29
>
> Dear reviewer,
>
> We thank you for your feedback and would like to use the chance to address your comments in the following sections.
>
> **Distinction between adding noise and RMLPs**. We would like to clarify a potential misunderstanding in the reviewer’s summary of our proposed method. Punctually, RMLPs are not implemented as fixed randomized MLPs that add Gaussian noise to intermediate features. Instead, as described in Section 3 and visualized in Figure 2, our approach involves multiplying the ViT’s output by a sequence of Gaussian matrices—i.e., matrices whose entries are independently drawn from a Gaussian distribution at each forward pass. This distinction is critical: the stochastic nature of the Gaussian matrices (with fresh samples at each call) prevents the model from overfitting to a static transformation, which would undermine the regularization effect. Moreover, unlike additive noise, the use of multiplicative random projections introduces specific structural properties to the representation space. For instance, the induced sparsity in token activations shown in Figure 6.a emerges as a direct consequence of this multiplicative mechanism. We believe this foundational difference —between additive perturbations and stochastic multiplicative transformations— plays a central role in f the representational and interpretability benefits observed in R$_\lambda$-ViT.
>
>
> **Further details on DINO re-implementation**. We confirm that our re-implementation of ViT was carefully validated by ensuring that it reproduced the same outputs as DINOv2-S when using pretrained weights, prior to any fine-tuning.
>
> **Omission of normalizations**. We thank the reviewer for their comment regarding the normalization layers in the DINO head. In our experiments, we compared downstream performance of both C-ViT and R$_\lambda$-ViT using DINO-style heads with and without the normalization layers, and observed no significant difference between the two configurations. Based on these findings, and in the interest of architectural simplicity, we chose to exclude them.
>
> We believe one contributing factor is that our models are fine-tuned from pretrained DINOv2 checkpoints, whereas the original DINO paper trains the vision transformer from scratch—making normalization more critical in the original context. Additionally, for R$_\lambda$​-ViT, Corollary 5 ensures that the token representations remain close to the unit sphere after passing through the randomized head. In the case of C-ViT, since the ViT backbone already produces well-structured token embeddings and the weights of the final linear layer are initialized within a bounded range, meaningful representations can still be learned in the absence of explicit normalization.
>
> That said, we acknowledge that we did not conduct a dedicated ablation to isolate the role of these normalization layers, and we appreciate the reviewer for emphasizing their relevance in the original DINO framework.
>
>
> **Resolution of attention maps and quantitative evaluation**. Our qualitative figures aimed to illustrate general alignment between attention distributions and semantically meaningful regions, not to serve as the primary evidence for interpretability and thus, were kept at low-res. The main support for our claims comes from statistical analyses (e.g., Figure 6).
> While we intentionally avoided high-resolution delineation of small structures to prevent over-interpretation, we acknowledge the importance of quantitative interpretability evidence. To that end, we conducted an additional experiment inspired by Darcet et al. (2024), using 1-NN classification on the OCT dataset from Eckardt et al. (2024). We compared performance when classifying patients with and without IRDs using the class token and patch tokens with the highest and lowest norms.
> Results show that R$_5$​-ViT performs well when using the highest-norm token, suggesting it attends to clinically relevant structures. In contrast, DINOv2-S performs the best with the lowest-norm token, indicating it may focus on anatomically void regions.
>
> |   Model   | Class Token | Highest-Norm Patch Token | Lowest-Norm Patch Token |
> |:-------------:|:---------------:|:-----------------------------:|:----------------------------:|
> |   DINOv2-S    |      0.76       |             0.48             |            0.76             |
> |  R$_5$-ViT    |      0.88       |             0.76             |            0.40             |
>
> Moreover, we evaluated the alignment between patch token norms and retinal layer annotations using the dataset introduced by Eckardt et al. (2024). To do this, we measured the correlation between the spatial positions of the top 25% highest-norm patch tokens from the models’ embeddings and the annotated retinal layer regions. Results below indicate no alignment with anatomically relevant structures for DINOv2-S, whereas R$_5$-ViT ($\lambda=5$) suggests improved localization of clinically relevant structures by our RMLP attention map. These findings provide quantitative support for our interpretability claims, particularly in the ophthalmological domain.
>
> |    Model    | Patients with IRD | Patients without IRD |
> |:-----------:|:-----------------:|:--------------------:|
> |  DINOv2-S   |       -0.06       |         0.09         |
> | R$_5$-ViT   |        0.36       |         0.33         |
>
> This supports our claim that R$_\lambda$-ViT produces more interpretable representations. We would be glad to include these quantitative results in the final manuscript.
>
> **Impact of KoLeo regularizer on training**. We politely disagree with the reviewer’s assessment that the KoLeo regularizer is unimportant in DINOv2. In DINOv2, Oquab et al. report clear performance gains in 1-NN classification and image retrieval when using KoLeo (Tables 1 and 3). In our work, KoLeo is not only retained for its empirical value, but also because it is mathematically essential for enabling Theorem 2, which assumes the injectivity of the ViT backbone.
>
> **Significance of theoretical results**. We respectfully disagree with the reviewer’s assessment regarding the theoretical contributions in Section 4. To the best of our knowledge, there is currently no existing mathematical formulation of ViTs that characterizes their representational and generalization properties in the manner presented in Theorems 1 and 2.
> We would also like to clarify that Theorem 4 and Corollary 5 go beyond illustrating that deterministic inputs become stochastic or that the involved noise should be small. They establish a rigorous connection between the dimensionality of the latent space and the magnitude of noise required to preserve topological properties, a relationship that is foundational for understanding the behavior of our RMLP regularizer, as discussed in lines 193–203 of the manuscript. These mathematical derivations provide a novel lens on how randomization and latent space geometry interact in contrastive learning setups. We appreciate the reviewer’s engagement with our theoretical analysis and hope this clarifies the motivation and significance of our formal contributions.
>
> **Manuscript’s formatting**. While we understand the reviewer’s concern regarding the manuscript structure, we would like to clarify that the current organization was a deliberate choice made in service of both clarity and depth. Although we do not include a standalone Related Work section, the Problem Formulation serves a similar role by reviewing key prior art relevant to our approach. More importantly, this structure allowed us to allocate sufficient space to our theoretical analysis, which we view as a central contribution of the paper. Our Theorems 1–4 offer a comprehensive mathematical foundation that explains how RMLP influences the geometry of contrastive learning—a contribution that would have been difficult to present in full had we followed a more conventional layout.
>
> **Including a formal introduction to DINO**. Given the widespread adoption and extensive documentation of DINO in the literature, we initially chose to reference and cite the original work rather than provide a detailed reintroduction in the main text. However, we agree that briefly summarizing the core components of DINO would improve the accessibility of the paper, especially for readers less familiar with self-supervised ViT training. We are happy to include a concise overview of DINO in the revised manuscript.
>
> **References**
>
> Darcet, T., Oquab, M., Mairal, J., Bojanowski, P.: Vision transformers need registers. arXiv. arXiv:2309.16588 (2024)
>
> Eckardt, F., Mittas, R., et al.: Deep Learning-Based Retinal Layer Segmentation in Optical Coherence Tomography Scans of Patients with Inherited Retinal Diseases. Klin Monbl Augenheilkd. doi: 10.1055/a-2227-3742 (2024)
>
> Oquab, M., Darcet, T., Moutakanni, T., Vo, H., Szafraniec, M., Khalidov, V., Fernandez, P., Haziza, D., Massa, F., El-Nouby, A., et al.: Dinov2: Learning robust visual features without supervision. arXiv preprint arXiv:2304.07193 (2023)

---

> > ### Comment · Reviewer_UqKd · 2025-08-05
> >
> > Thanks for the authors' response, and I have some followup questions:
> >
> > 1. I appreciate that the proposed framework is general and not limited to DINOv2, and I do not expect additional experimental results other than DINO during the discussion period. For methods that do not use KoLeo loss by default (e.g., DINOv1, iBOT, and CLIP), how do the authors plan to integrate the proposed framework? Moreover, since high-norm tokens are a general issue in Transformers, does the framework generalize beyond contrastively trained models?
> >
> > 2. A central aim of the paper appears to be enabling ViTs to produce more interpretable attention maps, particularly for medical applications. How do the authors envision these attention maps being used to guide real-world use cases? If the visualizations are deliberately presented at low resolution, how do the authors expect users to obtain improved interpretability from the proposed model?
> >
> > 3. I recognize that quantitatively assessing the interpretability of attention maps is challenging. A straightforward measure adopted by related work [a, b, c, d] is unsupervised object discovery or segmentation by applying STEGO [e] or LOST [f] directly to DINO’s attention maps. By contrast, the results in Figs. 3/4 do not necessarily reflect properties of the attention maps, as they are outputs of segmentation heads built on feature maps. Could the authors provide a quantitative comparison using the suggested protocol?
> >
> > 4. Darcet et al. [a] and some closely related ECCV’24 works, such as SINDER [b] and DVT [c], pursue similar goals but are absent from the experimental comparisons. Could the authors include them in both quantitative and qualitative evaluations?
> >
> > 5. Regarding the necessity of a dedicated related-work section, I believe there remains substantial room to help readers gain a fuller view of the topic. High-norm tokens (also referred to as massive tokens [h] or attention sinks [g]) and noisy attention maps [c] (or attention artifacts [i]) are common issues in Transformer-based architectures. The authors might also find the related-work section and experiments of the concurrent work [j] useful.
> >
> > References:
> >
> > [a] Darcet et al., Vision transformers need registers
> >
> > [b] Wang et al., SINDER: Repairing the Singular Defects of DINOv2
> >
> > [c] Yang et al., Denoising Vision Transformers
> >
> > [d] Helbling et al., ConceptAttention: Diffusion Transformers Learn Highly Interpretable Features
> >
> > [e] Hamiltion et al., Unsupervised Semantic Segmentation by Distilling Feature Correspondences
> >
> > [f] Simeoni et al., Localizing Objects with Self-Supervised Transformers and no Labels
> >
> > [g] Xiao et al., Efficient Streaming Language Models with Attention Sinks
> >
> > [h] Sun et al., Massive Activations in Large Language Models
> >
> > [i] Nakamura et al., Improving Image Clustering with Artifacts Attenuation via Inference-Time Attention Engineering
> >
> > [j] Jiang et al., Vision Transformers Don’t Need Trained Registers

---

> > > ### Author Response · Authors · 2025-08-08
> > > **Reply to follow up questions, part 1**
> > >
> > > Dear reviewer,
> > >
> > > we want to thank you very much for such a constructive feedback, relevant questions and for suggesting genuinely interesting experiments. We believe that incorporating these into the final version of our manuscript, as we intend to do, will significantly strengthen the paper’s impact and relevance.
> > > Below, we present the results of the experiments you proposed and address your questions, following the same numbering as in your most recent comment. We hope these additions and clarifications further demonstrate the significance of our approach.
> > >
> > > **Point 1.** While we understand that the reviewer did not expect additional experimental results, we nonetheless applied our method to MoCo-v3 [a] to further illustrate its generality. We specifically selected MoCo-v3 because it is a ViT model trained using a contrastive learning approach not belonging to the DINOv2 family and whose original training objective does not include the KoLeo regularizer.
> > >
> > > In this additional experiment, we fine-tuned MoCo-v3 on OCT data using the dataset from [b]. We then evaluated the model on disease classification (also on [b]) and semantic segmentation using the dataset from [c], following the same downstream protocols as in Table 2 of our paper.
> > >
> > > The results, shown below, indicate that our proposed regularization improves downstream performance even when applied to a model with a different training paradigm and latent dimensionality. This provides additional evidence that our framework is not specific to DINOv2, and can be beneficial across contrastively pretrained models, even when KoLeo is not part of the original training objective.
> > >
> > > That said, we emphasize—as stated in our manuscript (e.g., lines 29 and 85)—that our framework is designed with contrastive learning in mind. These new results reinforce this intended use and the versatility of our approach within this paradigm.
> > >
> > > **Accuracy on classification results on [b] including MoCov3**
> > >
> > > | Model                      | 1-NN   | Random Forest | Linear Classification |
> > > |:--------------------------:|:------:|:-------------:|:---------------------:|
> > > | MoCov3 + MLP               | 0.76   | 0.64          | 0.55                  |
> > > | MoCov3 + RMLP (λ=5)    | 0.76   | 0.66          | 0.59                  |
> > > | RetFound                   | 0.80   | 0.81          | 0.93 ± 0.05           |
> > > | DINOv2-S                   | 0.75   | 0.75          | 0.88 ± 0.08           |
> > > | R₅-ViT                     | 0.78 ± 0.01 | 0.74 ± 0.01 | 0.87 ± 0.01           |
> > >
> > > **Semantic segmentation results on [c] including MoCov3**
> > > | Model                        | Mean DICE (ViT-UNet hybrid) | DICE on ONL (ViT-UNet hybrid) |
> > > |-----------------------------|-----------------------------|-------------------------------|
> > > | MoCov3 + MLP                | 0.97                        | 0.69                          |
> > > | MoCov3 + RMLP (λ=5)    | 0.98                        | 0.73                          |
> > > | RetFound                    | 0.92 ± 0.06                 | 0.59 ± 0.10                   |
> > > | DINOv2-S                    | 0.79 ± 0.20                 | 0.54 ± 0.20                   |
> > > | R₅-ViT                      | 0.97 ± 0.03                 | 0.72 ± 0.09                   |
> > >
> > >
> > > **Point 2**.  A common use of AI in medical imaging is the localization of biomarkers [d, e]. With the rise of Vision Transformers (ViTs), recent works [f, g] have explored using attention maps to highlight disease-relevant regions. These maps are not used directly for diagnosis but serve to help clinicians understanding model decisions and building trust, particularly in complex or borderline cases.
> > >
> > >
> > > In our work, this is the intended use of attention maps. For instance, Figure 5 shows that after fine-tuning with RMLP, R$_5$-ViT learns to focus on meaningful areas for disease classification. We discuss this between lines 208–210, and further support it with new results in our rebuttal, where classification using only high- and low-norm patch tokens aligns with relevant image regions.
> > >
> > > While ViT attention maps can be coarse, in ophthalmology the diagnostic signals often span regions rather than fine structures—similar to observations in [g]. Nonetheless, we agree that higher-resolution maps would improve clarity and are happy to include improved visualizations in the final version. Unfortunately, we’re currently unable to upload images due to limitations of the review portal.

---

> > > > ### Author Response · Authors · 2025-08-08
> > > > **Reply to follow up questions, part 2**
> > > >
> > > > **Point 3** We thank the reviewer for the suggestion to include a STEGO-based quantitative evaluation of our attention maps. Due to time constraints, we provide results here only for the dataset from [c], which we selected given its difficulty for linear segmentation models, as shown in Tables 1 and 2.
> > > >
> > > > In the STEGO column of the table below, we report the mean IoU between expert annotations and predictions from 10 independently trained STEGO models [h], each performing binary segmentation (retinal layers vs. background). The low performance is consistent with prior work [i] and our own results in Table 2.b, likely due to STEGO’s use of a basic ReLU-activated feedforward segmentation head.
> > > > We will include comprehensive STEGO results across all models and datasets in the final version of the paper.
> > > >
> > > >
> > > > | Model      | STEGO          |
> > > > |:----------:|:--------------:|
> > > > | DINOv2-S   | 0.23 ± 0.06    |
> > > > | R$_5$-ViT  | 0.23 ± 0.05     |
> > > > | Registers  | 0.23 ± 0.06     |
> > > >
> > > > **Point 4.**  Due to time constraints during the discussion period, we report results only for [j] here. However, results for [k] and [l] will be included in the final version to broaden the comparison. For context, we also include relevant results from Tables 1 and 2 of our paper. We note that [g] was trained on hundreds of thousands of images and uses a large ViT, whereas our model was trained on only a few hundred images using a small ViT variant.
> > > >
> > > > Although no images are allowed to be uploaded in the current reviewer portal, we will also happily add the qualitative results from models [j,k,l] to the final version of the manuscript.
> > > >
> > > > **Accuracy on disease classification of [b]**
> > > > | Model        | Linear Classification | 1-NN | Random Forest |
> > > > |:------------:|:---------------------:|:----:|:-------------:|
> > > > | Registers [j] | 0.83                  | 0.76 | 0.68          |
> > > > | RetFound [g]  | 0.93                  | 0.80 | 0.81          |
> > > > | DINOv2-S     | 0.88                  | 0.75 | 0.75          |
> > > > | C-ViT        | 0.76                  | 0.59 | 0.51          |
> > > > | R$_5$-ViT    | 0.87                  | 0.78 | 0.74          |
> > > >
> > > > **Accuracy on disease classification of [m]**
> > > > | Model        | Linear Classification | 1-NN | Random Forest |
> > > > |:------------:|:---------------------:|:----:|:-------------:|
> > > > | Registers [j] | 0.74                  | 0.66 | 0.70          |
> > > > | RetFound [g]  | 0.83                  | 0.78 | 0.73          |
> > > > | DINOv2-S     | 0.79                  | 0.67 | 0.69          |
> > > > | C-ViT        | 0.71                  | 0.64 | 0.65          |
> > > > | R$_5$-ViT    | 0.75                  | 0.71 | 0.68          |
> > > >
> > > > **DICE scores on semantic segmentation of [c] using our ViT-UNet hybrid**
> > > > | Model         | Mean DICE | DICE on ONL |
> > > > |:-------------:|:---------:|:-----------:|
> > > > | Registers [j] | 0.96      | 0.64        |
> > > > | RetFound [g]  | 0.92      | 0.59        |
> > > > | DINOv2-S      | 0.79      | 0.54        |
> > > > | C-ViT         | 0.86      | 0.55        |
> > > > | R$_5$-ViT     | 0.97      | 0.72        |
> > > >
> > > > **RMSE on depth estimation of [n]**
> > > > | Model         | ViT-UNet hybrid | Linear head |
> > > > |:-------------:|:---------------:|:-----------:|
> > > > | Registers [j] | 8               | 10          |
> > > > | DINOv2-S      | 7               | 9           |
> > > > | C-ViT         | 8               | 10          |
> > > > | R$_5$-ViT     | 7               | 9           |
> > > >
> > > > **DICE score on semantic segmentation of [o]**
> > > > | Model         | ViT-UNet hybrid | Linear head |
> > > > |:-------------:|:---------------:|:-----------:|
> > > > | Registers [j] | 0.72            | 0.65        |
> > > > | DINOv2-S      | 0.81            | 0.76        |
> > > > | C-ViT         | 0.78            | 0.75        |
> > > > | R$_5$-ViT     | 0.8             | 0.77        |
> > > >
> > > > **Accuracy on image classification of [p]**
> > > > | Model         | Linear classification |
> > > > |:-------------:|:---------------------:|
> > > > | Registers [j] | 0.56                  |
> > > > | DINOv2-S      | 0.77                  |
> > > > | C-ViT         | 0.68                  |
> > > > | R$_5$-ViT     | 0.76                  |
> > > >
> > > > **Point 5.** We thank the reviewer for the helpful literature suggestions. As recommended, we will include a Related Work section in the final version to provide broader context on how high-norm tokens and noisy attention maps are known challenges in transformer architectures and give context to our work.

---

> > > > > ### Author Response · Authors · 2025-08-08
> > > > > **Reply to follow up questions, part 3**
> > > > >
> > > > > **References**
> > > > >
> > > > > [a] Chen, Xinlei, et al. Improved baselines with momentum contrastive learning
> > > > >
> > > > > [b] Gholami, P. et al.: OCTID: optical coherence tomography image database
> > > > >
> > > > > [c] Eckardt, F., Mittas, R., et al.: Deep Learning-Based Retinal Layer Segmentation in Optical Coherence Tomography Scans of Patients with Inherited Retinal Diseases
> > > > >
> > > > > [d] Freyre CAC et al., Biomarker-Based Classification and Localization of Renal Lesions Using Learned Representations of Histology-A Machine Learning Approach to Histopathology.
> > > > >
> > > > > [e] Asani, B. et al., Evaluation of OCT biomarker changes in treatment-naive neovascular AMD using a deep semantic segmentation algorithm
> > > > >
> > > > > [f] Wollek A, et al., Attention-based Saliency Maps Improve Interpretability of Pneumothorax Classification
> > > > >
> > > > > [g] Zhou, Y., A foundation model for generalizable disease detection from retinal images. Nature
> > > > >
> > > > > [h] Hamiltion et al., Unsupervised Semantic Segmentation by Distilling Feature Correspondences
> > > > >
> > > > > [i] Liu et al., Optimizing Vision Transformers for Medical Image Segmentation
> > > > >
> > > > > [j] Darcet et al., Vision transformers need registers
> > > > >
> > > > > [k] Wang et al., SINDER: Repairing the Singular Defects of DINOv2
> > > > >
> > > > > [l] Yang et al., Denoising Vision Transformers
> > > > >
> > > > > [m] Ahn, J. M. et al.: A deep learning model for the detection of both advanced and early glaucoma using fundus photography.
> > > > >
> > > > > [n] Nathan Silberman, Derek Hoiem, Pushmeet Kohli, and Rob Fergus. Indoor segmentation and support inference from rgbd images
> > > > >
> > > > > [o] Zhou, B. et al., Scene parsing through ade20k dataset
> > > > >
> > > > > [p] Russakovsky, O., et al., Imagenet large scale visual recognition challenge

---

> > > > > > ### Comment · Reviewer_UqKd · 2025-08-08
> > > > > >
> > > > > > Thanks for the authors' detailed response and I will raise my score to positive accordingly. Now I believe the manuscript could be revised to target a broader scope (contrastive-supervised ViTs) instead of DINOv2. I also hope that the authors could better align their demonstrations with target applications, try to perform evaluations on straightforward indicators for those tasks, incorporate comparisons with more closely related works, and connect with aforementioned background papers.

---

### Official Review · Reviewer_FxJh · 2025-07-05

**Clarity:** 3
**Significance:** 3
**Originality:** 3
**Rating:** 4
**Confidence:** 3

**Summary:**

The  basis of the manuscript is that the the authors state with quite a bit of evidence that ViT models trained with DINOv2 repurpose low-information patch tokens with global information.  This is particularly problematic for tasks that are segmenting in nature.  The specific domain of concern is  ophthalmologic images where the chief task is segmenting OCT and eye fundus images. The results also show improved performance on natural image datasets.

The solution proposed is to replace the MLP layer with a novel Randomized-Multi-Layer Perceptron (RMLP). RMLPs replace the standard learnable MLP heads in DINO and iBOT with fixed, randomized linear operators that preserve structure.  The authors believe this guards against shortcut learning by preventing token-level overfitting.  The goal is also to avoid major archetectural changes.

The results show qualitative improvements in depth estimation and segemenation on natural images.  Heatmaps on ophthalmologic images suggest that there is improved attention to the objects of interest and less attention attributed to the non-informative parts. This is most compelling in the OCT images. Quantitative results for segmentation and classification tasks are also provided in conclusions with some mixed results.  Most dramatic performance improvement is in "pathology classification" using Nearest Neighbor clustering.

**Questions:**

Please better define "probability balls". It is a confusing terminology and while I think I conceptually understand, it is hard to be certain.

I understand that you initialized with DINOv2-S trained on natural images and further trained on the different dataset after replacing the MLP heads.  Is there any advantage to this over training from scratch with RMLP only?

If the fundamental break through of the proposed algorithm is in improved segmentation, how does one understand the improved performance in classification task in Table 2a?

DINOv2 is the basis of many SOTA models in histopathologic image classification. Do you think that incorporating RMLP would improve performance in this domain?

**Ethical Concerns:**

["NO or VERY MINOR ethics concerns only"]

**Final Justification:**

Showing the change of the model induces attention to the biologically meaningful structures provides strong evidence that their simple model modification has solved what appears to be a valid problem with DINOv2 training.

**Limitations:**

Limitations section is perfunctory at best. This needs to be enhanced to be acceptable.

**Paper Formatting Concerns:**

Much of the results are in the discussion and there is very limited actual discussion.

**Quality:**

2

**Strengths And Weaknesses:**

Strengths:
Replacing MLP with RMLP might correct for hypothesis (with fair amount of evidence) that ViT models trained with DINOv2  store global information in low information patch tokens.
The theoretical proofs are very thorough and only have minor flaws.  I have some clarifying questions below

Weaknesses:
The heatmaps should have a legend to be considered interpretable. Simply stating "Brighter maps indicate more attention." is not adequate for full interpretation of the figures.
As a person not familiar with the domain of OCT and fundus image, I am not fully able to understand the goal of analyzing these image. Broadly defining these as "medical images" is also in appropriate because the goals for many other medical imaging tasks seem to be quite different from these ophthalmologic tasks.
Result section are largely qualitative which limits evaluation. Quantitative results are in the Conclusion section.

---

> ### Author Rebuttal · Authors · 2025-07-29
>
> Dear reviewer,
>
> We are glad that you found our theoretical thorough and appreciate that you recognized the potential of our method to tackle the norm-related artifacts from DINOv2. Following your constructive feedback, we would like to address your questions on the following sections.
>
> **Proper definition of “probability balls”**. A mathematical definition can be proposed as follows: Given a point $x \in \Omega$, $\|\cdot\|$ a norm  and a random matrix $\varphi$, we would say that $B(x, r) :=$ {$ z \in \Omega : \|z - x\| < r$ } is a probability ball if $\varphi x\in B(x,r)$ with high probability. In other words, a probability ball characterizes  the region in the latent space containing perturbations of a point under random linear transformations.. It reflects the typical spatial distortion induced by $\varphi$ and serves as a geometric intuition for analyzing local neighborhoods under randomized projections. We will be happy to include this definition in the final version of the paper to help formalize our terminology.
>
> **Fine tuning DINOv2 vs. training from scratch**. The numerical experiments we ran showed improved results on downstream tasks when fine-tuning DINOv2-S compared to training from scratch using RMLPs. This aligns with prior findings, such as Chettaoui et al. (2025), which similarly report that fine-tuning pretrained vision transformers tends to yield better performance than training from scratch. . Moreover, we decided to focus on low-resource setups (as commonly present in medical imaging), where both time and computational resources are limited. Initializing DINOv2 with weights pretrained on natural images has been shown to be beneficial for both of these factors while not falling short in performance metrics, like in Roth et al. (2024).
>
> **Improvement on classification tasks**. Although we observe strong performance improvements in segmentation across multiple modalities when using our proposed regularizer, we emphasize that these gains reflect broader enhancements in representation learning with R$_\lambda$-ViT, rather than isolated effects. By incorporating RMLPs, we promote more reliable and semantically robust image representations, which lead to consistent improvements across downstream tasks—most notably segmentation and classification. As shown in Table 2, R$_5$-ViT achieves better performance in 1-Nearest Neighbor (1-NN) classification than DINOv2-S, and comparable performance in random forest and linear classification settings.
>
> Notably, R$_5$-ViT also outperforms RetFound (Zhou et al., Nature, 2023)—a specialized foundation model trained on nearly a million of OCT scans—in both 1-NN classification (a proxy for image retrieval) and segmentation tasks. This result underscores the strong transfer learning potential of our method from natural images to medical imaging, despite being fine-tuned on a relatively small dataset of hundreds of OCT images.
>
> **Quantitative interpretation of attention maps**. We would like to begin by pointing out that in AI applied to ophthalmology, visual inspection remains a valuable tool to assess a model’s performance due to the challenging nature of interpretability in the field. Thus, similar to what RetFound (Zhou et at. Nature (2023)) did, the goal of Figure 5 is to show the different regions from the OCT and Fundus images that the considered models attend to, noticing how DINOv2-S focuses on void regions while R$_5$-ViT locates anatomically relevant structures such as retinal layers for the OCT or blood vessels in Fundus scans.
>
> To further support our interpretability claims, we conducted new experiments that quantitatively evaluate the alignment between model attention and anatomical structures, using the OCT dataset introduced by Eckardt et al. (2024), which includes patients with and without Inherited Retinal Disorders (IRDs). For each model, we identified the top 25% of patch tokens with the highest embedding norms and assessed how well their spatial positions aligned with the annotated retinal layer regions. As shown in the table below, DINOv2-S shows little to no alignment with anatomically meaningful structures, while R₅-ViT exhibits significantly improved localization, suggesting better semantic grounding in clinically relevant image regions.
>
> |    Model    | Patients with IRD | Patients without IRD |
> |:-----------:|:-----------------:|:--------------------:|
> |  DINOv2-S   |       -0.06       |         0.09         |
> | R$_5$-ViT   |        0.36       |         0.33         |
>
> We also conducted an experiment inspired by Darcet et al. (2024) to examine the type of information encoded in individual patch tokens. Using the dataset from Eckardt et al. (2024), we performed a classification task to distinguish patients with and without IRDs based on three inputs: the class token, the highest-norm patch token, and the lowest-norm patch token. The results, shown below, provide further quantitative evidence that R$_5$-ViT focuses on semantically meaningful, diagnostically relevant regions, whereas DINOv2-S appears to store classification-related information in less informative or void areas.
>
> |   Model   | Class Token | Highest-Norm Patch Token | Lowest-Norm Patch Token |
> |:-------------:|:---------------:|:-----------------------------:|:----------------------------:|
> |   DINOv2-S    |      0.76       |             0.48             |            0.76             |
> |  R$_5$-ViT    |      0.88       |             0.76             |            0.40             |
>
> We will be happy not only to make a more descriptive legend of Figure 5, but also to include these new experiments to make our manuscript more accessible to readers without familiarity with OCT or Fundus modalities.
>
> **Using RMLPs on histopathology**. Although we have not performed systematic experiments on histopathology images, the theoretical formulation of our method does not include any particularity from any modality. Given our experiments show improvements on classification tasks for microscopy images, it would be an exciting follow-up work to apply our regularizer to histopathology data, where improvements can be expected.
>
> **Used modalities**. We acknowledge that calling “medical images” to our ophthalmological data might be ambiguous. We will be happy to modify the manuscript to specify we restricted our experiments on retinal microscopy images into the ophthalmological field.
>
> **Position of quantitative results**. We will happily rearrange the paper to make sure that both quantitative and qualitative results are in the corresponding section.
>
> **Limitations**. We will be glad to extend our limitations’ section.
>
> **References**
>
> Chettaoui, T., Damer, N., Boutros, F. FRoundation: Are foundation models ready for face recognition? arXiv:2410.23831 (2025)
>
> Roth, B., Koch, V., Wagner, S., et al. Low-resource finetuning of foundation models beats state-of-the-art in histopathology. doi: 10.1109/ISBI56570.2024.10635695. (2024)
>
> Zhou, Y., Chia, M.A., Wagner, S.K. et al. A foundation model for generalizable disease detection from retinal images. Nature 622, 156–163 https://doi.org/10.1038/s41586-023-06555-x (2023)
> Eckardt, F., Mittas, R., et al.: Deep Learning-Based Retinal Layer Segmentation in Optical Coherence Tomography Scans of Patients with Inherited Retinal Diseases. Klin Monbl Augenheilkd. doi: 10.1055/a-2227-3742 (2024)
>
> Darcet, T., Oquab, M., Mairal, J., Bojanowski, P.: Vision transformers need registers. arXiv. arXiv:2309.16588 (2024)

---

> > ### Comment · Reviewer_FxJh · 2025-08-06
> > **Raise assessment to bordeline accept**
> >
> > The authors have provided the rebuttal which I believe significantly elevates their contribution.  Providing quantitive results as to the improved focus of the model on anatomically meaningful structures  helps to elevate the results to a presentable manuscript.

---

### Official Review · Reviewer_wDm7 · 2025-07-07

**Clarity:** 3
**Significance:** 3
**Originality:** 4
**Rating:** 3
**Confidence:** 3

**Summary:**

This paper proposes Randomized-MLP(RMLP) to improve domain adaptation interpretability and reduce attention map artifacts in DINOv2 by replacing learnable MLP heads with randomized, non-trainable operators. The method aims to address artifacts where DINOv2-S repurposes low-informative patches for global information storage, particularly problematic in medical imaging. The work combines theoretical analysis with empirical validation across  both natural and medical image domains.

**Questions:**

(1)Do you have a principled method for selecting λ for new domains, or is manual tuning required?

(2) Could RMLP be implemented on other self-supervised ViTs or the method is specific to DINOv2's architecture?

**Ethical Concerns:**

["NO or VERY MINOR ethics concerns only"]

**Limitations:**

The authors have properly addressed its limitations.

**Quality:**

3

**Strengths And Weaknesses:**

Strength:

(1) Lightweight component: RMLP requires no architecture modifications and trainable parameters that can be applied during fine-tuning with minimal computational overhead. The method is theoretically motivated and simple to implement.
(2) Solid theoretical framework: Theorems 1-4 provide comprehensive mathematical analysis of how RMLP regularization influences contrastive learning geometry, offering valuable insights into the mechanism underlying the observed improvements.
(3) Comprehensive experiments: the experiments show multiple domains and tasks such as classification, segmentation and depth estimation with proper statistical significance testing using Mann-Whitney U tests.

Weakness:

(1) The paper shows experiment results only on DINOv2-S based models, experiments on other relevant models such as DINOv2-base and large are untested.
(2) Apart from Figure 6,this paper relies heavily on visual results to show the enhanced interpretability of attention maps. Providing more quantitative evaluation results would strengthen the claims.
(3) RMLP shows performance degradation on some downstream tasks, indicating the method seems to sacrifice linear separability for neighborhood structure.

---

> ### Author Rebuttal · Authors · 2025-07-29
>
> Dear reviewer,
>
> We are very glad to read that you find our theoretical framework solid and our method properly evaluated across modalities and tasks. We also value very much that you find our work highly original and appreciate the constructive feedback. In the following sections, we would like to address your remaining questions.
>
> **Principled method for selecting** $\lambda$. While we do not yet provide an automated procedure for determining $\lambda$, we offer both an intuitive visualization and a theoretical foundation to guide its choice. As shown in Figure 2, if $\lambda$ is too small, the resulting permutations are insufficient to act as an effective regularizer; if too large, $\lambda$ can distort the embedding topology, negatively impacting performance. To support this principle, Theorem 4 establishes a mathematical connection between $\lambda$ and the dimensionality of the latent space, offering a theoretical basis that may inform more systematic approaches in the future. In our experiments, we selected $\lambda$ via manual grid search. We evaluated performance across various downstream tasks and image modalities, consistently finding that $\lambda =5$ yielded the best results. While this suggests a possible heuristic, we acknowledge that further empirical validation across a wider range of datasets and architectures is needed to establish a generalizable selection method.
>
> **RMLP implementation on other architectures**. Our theoretical analysis suggests that RMLPs can be applied broadly to any contrastive learning setup, independent of the underlying ViT architecture. To support this, we conducted some preliminary tests using the DINOv2 base model and observed a similar performance improvement by RMLPs on downstream tasks. Nevertheless, we decided not to include these results in this manuscript’s final version because we wanted to focus on the low-resource setup, to improve computational efficiency in the medical imaging field. Exciting follow-up projects could include investigating the effects of RMLPs not only on architectures like MoCo and Swin Transformers used in vision tasks, but also on those used for natural language processing, such as SimSCE or BERT.
>
> **Qualitative assessment in ophthalmological models**. We fully agree that interpretability is a critical aspect in understanding model behavior. At the same time, we would like to emphasize that in ophthalmology, quantifying interpretability remains challenging. As highlighted in RetFound (Zhou et al. Nature (2023)) and other prior work (e.g., Eckardt et al. (2024)), interpretability in this domain is often best evaluated through spatial correlation with anatomical structures and clinical relevance as judged by domain experts, rather than through scalar metrics alone. Nevertheless, following the reviewer's suggestion, we also conducted additional quantitative analysis on a dataset where ground-truth annotations of anatomical structures are available (see below), to complement our qualitative analysis.
>
> **New quantitative results (Correlating high-norm patches and anatomical structures)**. We recognize the importance of complementing qualitative insights with quantitative analysis and have therefore conducted additional experiments to better support our interpretability claims. Precisely, we evaluated the alignment between patch token norms and retinal layer annotations using the dataset introduced by Eckardt et al. (2024), which includes OCT scans with expert-labeled retinal layers from patients with and without Inherited Retinal Disorders (IRDs). To do this, we measured the correlation between the spatial positions of the top 25% highest-norm patch tokens from the models’ embeddings and the annotated retinal layer regions. Results below indicate no alignment with anatomically relevant structures for DINOv2-S, whereas R$_5$-ViT ($\lambda=5$) suggests improved localization of clinically relevant structures by our RMLP attention map. These findings provide quantitative support for our interpretability claims, particularly in the ophthalmological domain.
>
> |    Model    | Patients with IRD | Patients without IRD |
> |:-----------:|:-----------------:|:--------------------:|
> |  DINOv2-S   |       -0.06       |         0.09         |
> | R$_5$-ViT   |        0.36       |         0.33         |
>
> **New quantitative results (Information in patch tokens)**. We further investigated model behavior by analyzing the diagnostic information contained in individual patch tokens. Following a protocol similar to Darcet et al. (2024), we performed a 1-nearest neighbor classification task to distinguish patients with and without IRDs from the Eckardt et al. (2024) dataset. Specifically, we compared classification accuracy when using: (1) the class token, (2) the highest-norm patch token, and (3) the lowest-norm patch token.
>
> The results (see table below) reveal that while DINOv2-S achieves reasonable accuracy using the class token (0.76), its highest-norm patch token performs poorly (0.48)—even worse than the lowest-norm token (0.76). This suggests that DINOv2-S attends to non-specific or potentially irrelevant regions. In contrast, R$_5$-ViT maintains high class token accuracy (0.88) and achieves strong performance (0.76) using only the highest-norm patch token, while the lowest-norm token performs poorly (0.40), as expected. These findings support our claim that RMLP encourages attention to diagnostically meaningful areas, rather than repurposing void regions.
>
>
> |   Model   | Class Token | Highest-Norm Patch Token | Lowest-Norm Patch Token |
> |:-------------:|:---------------:|:-----------------------------:|:----------------------------:|
> |   DINOv2-S    |      0.76       |             0.48             |            0.76             |
> |  R$_5$-ViT    |      0.88       |             0.76             |            0.40             |
>
> **Additional comments on expert evaluation**. We would like to emphasize that our approach to interpretability extends beyond raw attention visualizations in two important ways. First, in Figure 5 and in Appendix Figures 15, 17, and 19, we present qualitative examples where R-ViT models attend to disease-relevant anatomical structures, whereas DINOv2-S does not. These examples were carefully reviewed by ophthalmologists and serve as task-relevant, clinically informed evidence of improved interpretability. Second, we provide a statistical analysis of attention map distributions in Figure 6. For example, Subfigure 6b illustrates how the norm distributions shift when comparing low- and high-information patches—capturing structure-aware changes that reflect the semantic properties of the image modality. These analyses go beyond visual intuition, offering quantitative and anatomically grounded insights into how model attention behavior changes with RMLP.
>
>
> **Linear separability vs. neighborhood-based classification**. We would like to clarify that, on average (as reported in Table 2a. Look at table below), DINOv2-S and R₅-ViT perform equivalently on linear classification tasks, while R$_5$-ViT consistently outperforms DINOv2-S in 1-Nearest Neighbor classification. This distinction aligns with our design: as discussed in lines 193–203, MLP heads in DINOv2-S tend to promote global linear separability, whereas our randomized MLPs (RMLPs) encourage local neighborhood preservation in the latent space—a hallmark of contrastive learning. Therefore, the performance differences observed in specific tasks reflect the intended trade-off between linear discriminability and semantic locality, with RMLPs offering advantages in tasks that benefit from structured, neighborhood-aware representations, like 1-NN classification.
>
> Notably, R₅-ViT outperforms RetFound (Zhou et al. Nature (2023), a specialized retinal foundation model trained on hundreds of thousands of OCT scans, in both 1-NN classification (can be used for image retrieval) and segmentation tasks (See Table 2b). This demonstrates the strong transfer learning potential of our method from natural images to medical images, even when fine-tuned on a relatively small dataset of hundreds of OCT images.
>
> |   Model   | 1-Nearest Neighbor | Random Forest | Linear classification |
> |:-------------:|:---------------:|:-----------------------------:|:----------------------------:|
> |   DINOv2-S    |      0.60       |            0.58             |            0.64 $\pm$ 0.06             |
> |  R$_5$-ViT    |      0.64 $\pm$ 0.02      |             0.58 $\pm$ 0.03             |            0.64 $\pm$ 0.02             |
>
> **References**
>
> ​​Eckardt, F., Mittas, R., et al.: Deep Learning-Based Retinal Layer Segmentation in Optical Coherence Tomography Scans of Patients with Inherited Retinal Diseases. Klin Monbl Augenheilkd. doi: 10.1055/a-2227-3742 (2024)
>
> Zhou, Y., Chia, M.A., Wagner, S.K. et al. A foundation model for generalizable disease detection from retinal images. Nature 622, 156–163 https://doi.org/10.1038/s41586-023-06555-x (2023)
>
> Darcet, T., Oquab, M., Mairal, J., Bojanowski, P.: Vision transformers need registers. arXiv. arXiv:2309.16588 (2024)

---

> > ### Author Response · Authors · 2025-08-07
> >
> > Dear Reviewer,
> >
> > Thank you once again for your constructive feedback and positive comments. We hope our rebuttal has adequately addressed your questions. Additionally, as part of our discussion with Reviewer WhNa, we have included preliminary results using a non-DINOv2 model, which we believe addresses your concerns on this matter.
> >
> > We look forward to your further feedback and are happy to answer any additional questions you may have.
> >
> > Sincerely,
> > The authors of Submission 15818

---

> > > ### Author Response · Authors · 2025-08-08
> > >
> > > Dear reviewer,
> > >
> > > We are writing to you to address your concern about our method only being tested on DINOv2-S model and to share with you the new experiments we ran using MoCo-v3 [a] to further illustrate its generality. We specifically selected MoCo-v3 because it is a ViT model trained using a contrastive learning approach not belonging to the DINOv2 family and whose original training objective does not include the KoLeo regularizer, a fundamental piece on our analysis.
> > >
> > > In this additional experiment, we fine-tuned MoCo-v3 on OCT data using the dataset from [b]. We then evaluated the model on disease classification (also on [b]) and semantic segmentation using the dataset from [c], following the same downstream protocols as in Table 2 of our paper.
> > >
> > > The results, shown below, indicate that our proposed regularization improves downstream performance even when applied to a model with a different training paradigm and latent dimensionality. This provides additional evidence that our framework is not specific to DINOv2, and can be beneficial across contrastively pretrained models.
> > >
> > > **Semantic segmentation results including MoCov3**
> > >
> > > | Model                    | Mean DICE ViT-UNet hybrid | DICE on ONL ViT-UNet hybrid |
> > > |:------------------------:|:-------------------------:|:---------------------------:|
> > > | MoCov3+MLP               | 0.97                      | 0.69                        |
> > > | MoCov3+RMLP ($\lambda=5$) | 0.98                      | 0.73                        |
> > > | RetFound                 | 0.92$\pm$0.06             | 0.59$\pm$0.1                |
> > > | DINOv2-S                 | 0.79$\pm$0.2              | 0.54$\pm$0.2                |
> > > | C-ViT                    | 0.86$\pm$0.2              | 0.55$\pm$0.2                |
> > > | R$_5$-ViT                | 0.97$\pm$0.03             | 0.72$\pm$0.09               |
> > >
> > >
> > > **Accuracy on classification results including MoCov3**
> > >
> > > | Model                      | 1-NN          | Random Forest   | Linear classification    |
> > > |:--------------------------:|:-------------:|:---------------:|:------------------------:|
> > > | MoCov3 + MLP               | 0.76          | 0.64            | 0.55                     |
> > > | MoCov3 + RMLP ($\lambda=5$) | 0.76          | 0.66            | 0.59                     |
> > > | RetFound                   | 0.80          | 0.81            | 0.93$\pm$0.05            |
> > > | DINOv2-S                   | 0.75          | 0.75            | 0.88$\pm$0.08            |
> > > | C-ViT                      | 0.59$\pm$0.02 | 0.51$\pm$0.01   | 0.76$\pm$0.02            |
> > > | R$_5$-ViT                  | 0.78$\pm$0.01 | 0.74$\pm$0.01   | 0.87$\pm$0.01            |
> > >
> > > **References**
> > >
> > > [a] Chen, Xinlei, et al. Improved baselines with momentum contrastive learning
> > >
> > > [b] Gholami, P. et al.: OCTID: optical coherence tomography image database
> > >
> > > [c] Eckardt, F., Mittas, R., et al.: Deep Learning-Based Retinal Layer Segmentation in Optical Coherence Tomography Scans of Patients with Inherited Retinal Diseases

---

### Decision · Program_Chairs · 2025-09-17

**Decision:**

Accept (poster)

**Comment:**

This paper aims to improve the domain adaptation interpretability of the attention/feature maps of DINO, and applies the approach for medical image analysis. The manuscript is well written and clearly structured. The rebuttal process was effective. The authors successfully addressed major concerns of some reviewers by providing detailed explanations and additional evaluation with quantitative results. This paper overall received reviewers' ratings of borderline reject, borderline accept, borderline accept, accept. The AC takes the majority votes and would like to recommend a final acceptance of this submission.